# DYRK4 upregulates antiviral innate immunity by promoting IRF3 activation

Xianhuang Zeng[1], Jiaqi Xu[1], Jiaqi Liu[2], Yang Liu[1], Siqi Yang[1], Junsong Huang[2], Chengpeng Fan[1], Mingxiong Guo [2,3✉] & Guihong Sun [1,4✉]

## Abstract

Viral infection activates the transcription factors IRF3 and NF-κB, which induce type I interferon (IFN) and antiviral innate immune responses. Here, we identify dual-specific tyrosine phosphorylation-regulated kinase 4 (DYRK4) as an important regulator of virus-triggered IFN-β induction and antiviral innate immunity. Overexpression of DYRK4 enhances virus-triggered activation of IRF3 and type I IFN induction, whereas knockdown or knockout of DYRK4 impairs virus-induced activation of IRF3 and NF-κB. Moreover, *Dyrk4*-knockout mice are more susceptible to viral infection. The underlying mechanism involves DYRK4 acting as a scaffold protein to recruit TRIM71 and LUBAC to IRF3, increasing IRF3 linear ubiquitination, maintaining IRF3 stability and activation during viral infection, and promoting the IRF3-mediated antiviral response. Our findings provide new insights into the molecular mechanisms underlying viral infection-triggered IRF3 stabilization and activation.

**Keywords** Antiviral Innate Immunity; DYRK4; IRF3; TRIM71
**Subject Categories** Immunology; Microbiology, Virology & Host Pathogen Interaction; Post-translational Modifications & Proteolysis

## Introduction

The innate immune system is the host's first line of defense against infection by pathogenic microorganisms (Kasuga et al, 2021). Following viral infection, pathogen-associated molecular patterns (PAMPs) are recognized by pattern recognition receptors (PRRs), including Toll-like receptors (TLRs), RIG-I-like receptors (RLRs), NOD-like receptors (NLRs), C-type receptors (CLRs), AIM2-like receptors (ALRs) and cytoplasmic DNA sensors (cGAS), and initiate innate antiviral immune responses by detecting viral RNA or DNA (Galluzzi et al, 2018; Li and Wu, 2021). After recognizing virus-specific features, PRRs hierarchically trigger signaling cascades, including the activation of downstream adaptor proteins

(TRAF3, MYD88, MAVS, and STING) (Seth et al, 2005; Ishikawa et al, 2009), serine/threonine kinases (TBK1 and IKBKE) and several transcription factors (NF-κB, IRF3, and IRF7) (Chan and Gack, 2016), to initiate the transcription of type I interferon (IFN) and proinflammatory cytokine genes, thus establishing the innate immune state against invading microorganisms (Chen et al, 2016; Rehwinkel and Gack, 2020).

IRF3 is a key transcription factor required for the antiviral innate immune response. After receiving the upstream TBK1 signal, IRF3 activates type I IFN signaling through multiple processes, including phosphorylation, dimerization, translocation to the nucleus, and stimulation of DNA binding and transcriptional activity of type I IFN genes (Taniguchi et al, 2001). Its activity and stability are regulated by multiple posttranslational modifications (PTMs) mediated by phosphorylation, ubiquitination, and sumoylation. The C-terminal phosphorylation cluster mutants IRF3-2A or IRF3-5A impair its activation (Lin et al, 2023). Furthermore, linear ubiquitination of IRF3 in RNA virus-infected cells activates the RLR-induced IRF3-mediated apoptotic pathway (RIPA), and IRF3 acts as a proapoptotic factor that protects the host in the absence of antiviral gene induction (Chattopadhyay et al, 2016). The ubiquitin E3 ligases RAUL, c-Cbl, RBCK1, TRIM26, UBE3C, and other proteins PIN1 and FOXO1 affect the ubiquitination of IRF3 and promote its degradation (Zhao et al, 2016; Saitoh et al, 2006; Lei et al, 2013; Zhang et al, 2008; Yu and Hayward, 2010; Fujita et al, 2015), whereas USP14 negatively affects IRF3 activation by removing the k63-linked ubiquitination of IRF3 (Wu et al, 2022), and RNF138 negatively regulates virus-triggered signaling by inhibiting the interaction of PTEN with IRF3 (Zeng et al, 2023). In contrast, TRIM21 prevents IRF3 ubiquitination and degradation and maintains IRF3 stability by interfering with the interaction between Pin1 and IRF3 (Yang et al, 2009). In virus-infected cells, Ubc5 is an E2-conjugating enzyme required for K63-linked ubiquitination resulting from IRF3 activation (Zeng et al, 2009). In addition, the linear ubiquitin chain assembly complex (LUBAC) mediates the linear ubiquitination of IRF3, which triggers RIPA (Chattopadhyay et al, 2016). However, Otulin inhibits RIPA by deubiquitinating IRF3 (Raja and Sen, 2021). Recent studies have shown that the deubiquitinases BAP1, OTUD7B, and PSMD14 regulate IRF3 levels and activity through ubiquitination or ubiquitination–autophagy pathways to modulate antiviral

[1]Taikang Medical School (School of Basic Medical Sciences), Wuhan University, 430071 Wuhan, China. [2]Hubei Key Laboratory of Cell Homeostasis, College of Life Sciences, Wuhan University, 430072 Wuhan, China. [3]School of Ecology and Environment, Tibet University, 850000 Lhasa, Xizang, China. [4]Hubei Provincial Key Laboratory of Allergy and Immunology, 430071 Wuhan, China. ✉E-mail: guomx@whu.edu.cn; ghsunlab@whu.edu.cn

immunity (Xie et al, 2022; Wu et al, 2020; Liu et al, 2024). The methyltransferase NSUN2 catalyzes the m5C modification of IRF3 mRNA, which results in its degradation and reduces protein levels (Wang et al, 2023). However, the detailed mechanisms that positively regulate the activity and stability of IRF3 remain poorly understood.

DYRK4 is a member of the conserved dual-specific tyrosine phosphorylation-regulated kinase (DYRK) family, which contains five members (DYRK1A, DYRK1B, DYRK2, DYRK3, and DYRK4) and has a conserved N-terminal DYRK homology cassette (DH) and an adjacent kinase domain (Aranda et al, 2011). It has a highly conserved Tyr-X-Tyr amino acid motif in the catalytic domain of the activation loop, and phosphorylation of the second tyrosine residue is essential for full activation of DYRKs, with mature DYRKs phosphorylating only serine or threonine residues on the substrate (Kinstrie et al, 2010; Lochhead et al, 2005). Numerous studies have shown that the DYRK family plays a key role in human disease, as DYRK1A/B and DYRK2/3 are involved in various signaling pathways essential for cellular homeostasis and developmental processes (Abbassi et al, 2015; Friedman, 2007; Nihira and Yoshida, 2015; Ma et al, 2019). During signaling pathways, DYRK3 also plays a role in regulating the stability of P granule-like structures (Wippich et al, 2013), and DYRK2 functions as a scaffold for forming the E3 ubiquitin ligase EDVP complex and as a kinase that phosphorylates ligase substrates (Maddika and Chen, 2009). In contrast to other DYRKs, little is known about the function of DYRK4, and no substrate has been identified for this kinase. It has been reported that mouse and rat *Dyrk4* is a testis-specific kinase predominantly expressed in the testis. However, *Dyrk4*-deficient mice are fertile (Sacher et al, 2007), which may be redundant in terms of function because DYRKs are strongly expressed in the testis. A tumorigenic chimeric transcript, RAD51AP1-DYRK4, enhances the activation of MEK/ERK signaling and increases the invasiveness of ductal mammary carcinomas (Liu et al, 2021) and hypomethylation of DYRK4 in peripheral blood is associated with increased lung cancer risk (Qiao et al, 2023). Recently, DYRK2 was found to negatively regulate antiviral innate immunity by promoting TBK1 degradation through phosphorylation (An et al, 2015). However, whether and how DYRK4 is involved in virus-triggered signaling was so far not known.

This study shows that DYRK4 is essential for virus-triggered IRF3 and NF-κB activation, IFNβ induction, and cellular antiviral responses. DYRK4 acts as a scaffold protein to recruit TRIM71 to interact with IRF3, increasing IRF3 linear ubiquitination, maintaining IRF3 stability during viral infection, and promoting IRF3-mediated antiviral responses. Our findings provide insight into the molecular mechanisms underlying virus-triggered IRF3 activation and stabilization, a critical step in virus-triggered type I IFN induction and cellular antiviral responses.

# Results

## DYRK4 positively regulates virus-triggered signaling

Given that DYRK4's cognate family member, DYRK2, is involved in the regulation of innate immune responses to RNA and DNA viruses (An et al, 2015), we investigated whether DYRK4 regulates innate immune responses to RNA and DNA viruses. We first

analyzed the dynamic expression of DYRK4 in THP-1 cells. We found that the mRNA and protein levels of DYRK4 were increased under SeV, VSV, or HSV-1 infection conditions (Fig. EV1A), suggesting a potential link between DYRK4 and antiviral immune responses. Next, we examined whether the upregulation of DYRK4 was involved in the antiviral innate immune response. Reporter assays revealed that DYRK4 overexpression enhanced SeV-induced activation of the IFNβ, ISRE, and NF-κB promoters (Fig. EV1B) in a dose-dependent manner (Fig. 1A). The overexpression of DYRK4 increased the SeV-induced transcription of downstream genes, including IFNB1, ISG15, CXCL10, and IL-6 (Fig. EV1C). Furthermore, overexpression of DYRK4 enhanced SeV-induced phosphorylation of IRF3 in the IRF3 pathway and phosphorylation of IκBα and p65 in the NF-κB pathway (Fig. EV1D), respectively, and enhanced SeV-induced dimerization of IRF3 (Fig. EV1E) and nuclear translocation of IRF3 (Fig. EV1F), which are the hallmarks of IRF3 activation. Furthermore, plaque assays revealed that DYRK4 overexpression significantly inhibited VSV-GFP replication (Fig. EV1G). These results indicate that DYRK4 may be involved in the virus-triggered activation of IRF3 and the transcription of downstream antiviral genes.

We next examined whether endogenous DYRK4 is required for virus-triggered IRF3 activation. DYRK4-deficient HEK293T and DYRK4-knockdown A549 cells were generated via CRISPR-Cas9 system (Fig. EV1H). Reporter assays revealed that SeV-induced activation of the IFN-β promoter was inhibited in DYRK4-deficient HEK293T cells (Fig. 1B). Compared with those in control cells, the transcription of downstream genes such as IFNB1, ISG15, IL-6, and CXCL10 induced by SeV was significantly inhibited in DYRK4-deficient HEK293T and DYRK4-knockdown A549 clones (Figs. 1C and EV1I), respectively. Consistently, HSV-1-induced transcription of the downstream gene IFNB1 was significantly inhibited in the DYRK4-knockdown A549 cells (Fig. 1D). In addition, the SeV-induced phosphorylation of IRF3, p65, and IκBα was significantly lower in DYRK4-deficient HEK293T and DYRK4-knockdown A549 cells than in control cells (Figs. 1E and EV1J). Similar results were obtained for IRF3, p65, and IκBα phosphorylation in HSV-1-infected DYRK4-knockdown A549 cells (Fig. 1F). Furthermore, SeV-induced IRF3 dimerization (Fig. 1G) and nuclear translocation of IRF3 (Fig. 1H) were inhibited in DYRK4-deficient HEK293T-cells, respectively. Fluorescence microscopic analysis of A549 cells revealed that VSV-GFP replication was significantly promoted in DYRK4-knockdown A549 cells compared with control cells (Fig. 1I). These results suggest that DYRK4 is required for virus-triggered signaling in different cell types.

To independently confirm the function of DYRK4 in virus-triggered signaling in humans, we constructed two human DYRK4 RNAi plasmids to test in human cells. Reporter assays revealed that the knockdown of DYRK4 inhibited SeV-induced activation of the IFN-β promoter, ISRE, and NF-κB (Fig. EV1K) and inhibited SeV-induced transcription of downstream genes such as IFNB1, ISG15, IL-6, and CXCL10 (Fig. EV1L). In addition, the SeV-induced phosphorylation of IRF3 and p65 was inhibited by DYRK4 knockdown in HEK293T cells (Fig. EV1M). These results confirm that DYRK4 plays an important role in the RLR-mediated signaling pathway.

Previous studies have shown that the kinase activity of DYRKs has important functions (An et al, 2015; Aranda et al, 2011); thus, we examined whether the effect of DYRK4 on virus-triggered IRF3

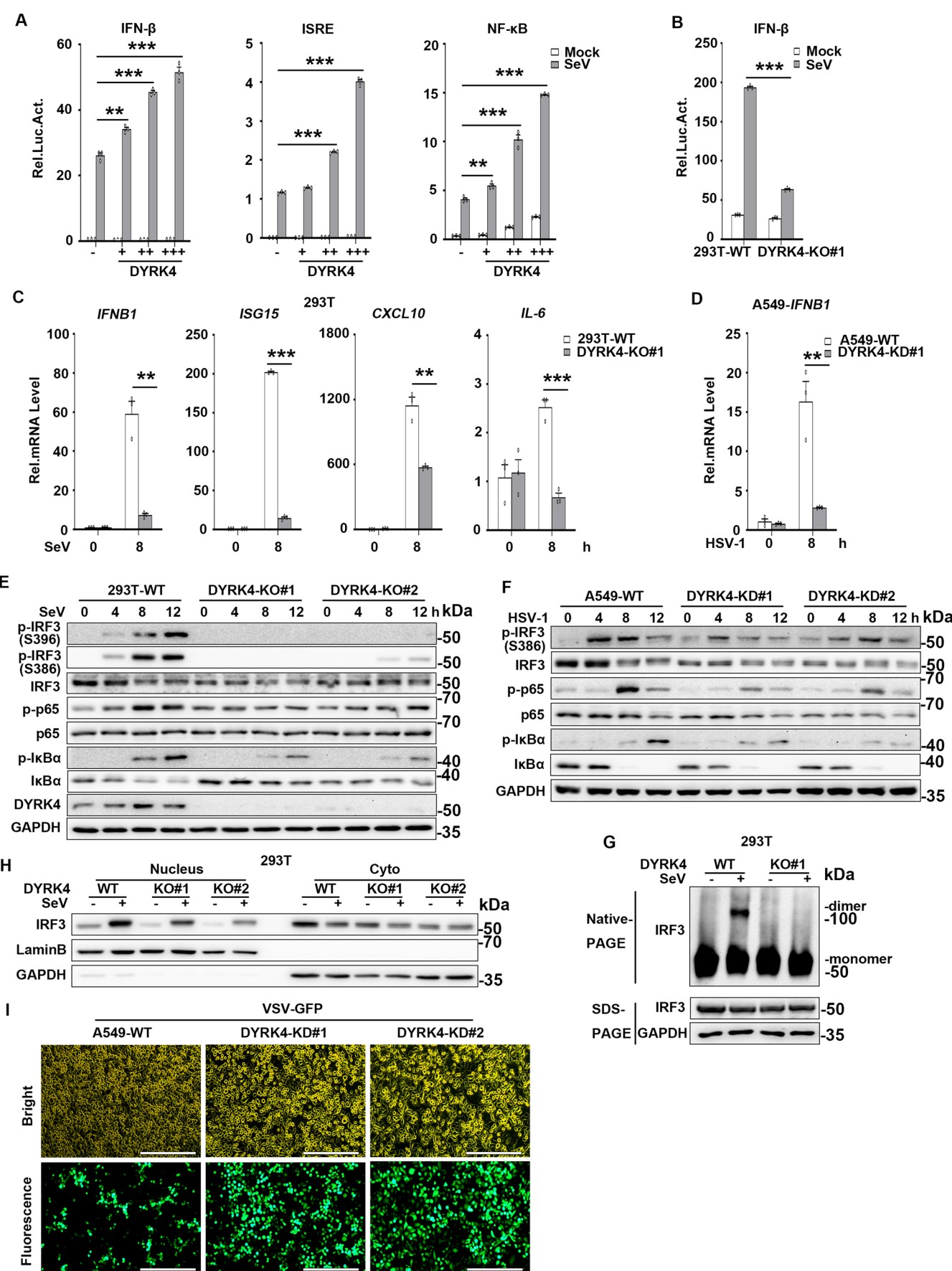

**Figure 1.  Identification of DYRK4 as a positive regulator of virus-triggered signaling.**

(A) DYRK4 activated the IFN-β promoter, ISRE, and NF-κB in a dose-dependent manner. HEK293T cells were co-transfected with the IFN-β, ISRE, and NF-κB reporters and increased amounts of the DYRK4 plasmid for 24 h and then infected with SeV for 12 h before luciferase assays. (P value; IFN-β; DYRK4 + : 0.0011, DYRK4 + + : 3.5 × 10$^{-5}$, DYRK4 + + + : 1.3 × 10$^{-4}$; ISRE; DYRK4 + + : 1.5 × 10$^{-6}$, DYRK4 + + + : 8.3 × 10$^{-7}$; NF-κB; DYRK4 + : 0.003, DYRK4 + + : 0.0002, DYRK4 + + + : 2.0 × 10$^{-7}$) (n = 3 biological replicates). (B) Effects of DYRK4 deficiency on SeV-induced activation of the IFN-β promoter. DYRK4-KO and control HEK293T cells were infected with SeV for 8 h before luciferase assays. (P value = 1.3 × 10$^{-7}$) (n = 3 biological replicates). (C) Effects of DYRK4 deficiency on the SeV-induced transcription of downstream genes. DYRK4-KO and control HEK293T cells were infected with SeV for 8 h before qPCR analysis. (P value; IFNB1: 0.0013, ISG15: 4.0 × 10$^{-8}$, CXCL10: 0.0016, IL-6: 0.0004) (n = 3 biological replicates). (D) Effects of DYRK4 knockdown on the HSV-1-induced transcription of downstream genes. DYRK4-KD and control A549 cells were infected with HSV-1 for 8 h before qPCR analysis. (P value = 0.0062) (n = 3 biological replicates). (E) Effects of DYRK4 deficiency on SeV-induced phosphorylation of IRF3 (Ser386, Ser396), p65, and IκBα. DYRK4-KO and control HEK293T cells were infected with SeV for the indicated times before immunoblotting analysis with the indicated antibodies. (F) Effects of DYRK4 knockdown on HSV-1-induced phosphorylation of IRF3 (Ser386), p65 and IκBα. DYRK4-KD and control A549 cells were infected with HSV-1 for the indicated times before immunoblotting analysis with the indicated antibodies. (G) Effects of DYRK4 deficiency on SeV-induced dimerization of IRF3. DYRK4-KO and control HEK293T cells were infected with SeV for 8 h, after which the cell lysates were separated by native (upper panel) or SDS (bottom panel) PAGE and analyzed by immunoblotting with the indicated antibodies. (H) Effects of DYRK4 deficiency on the SeV-induced nuclear translocation of IRF3. DYRK4-KO and control HEK293T cells were infected with SeV for 8 h, and then immunoblot analysis of IRF3 in the cytoplasmic (Cyto) and nuclear fractions was performed with the indicated antibodies. (I) Effects of DYRK4 knockdown on VSV-GFP replication. DYRK4-KD and control A549 cells were infected with VSV-GFP (MOI (multiplicity of infection) of 0.1) for 12 h before fluorescence microscopy analysis. (Bright field, upper panel; fluorescence, bottom panel). Scale bars, 100 μm. Data information: Data are representative of three biological replicates and are shown as the means with SEMs (A–D); data in (E–I) are representative of two replicates. **P < 0.01, ***P < 0.001; two-tailed unpaired Student's t test. Source data are available online for this figure.

activation depends on its kinase activity. We constructed three DYRK4 mutants (Papadopoulos et al, 2011), a 133-site mutant (K133R or K133A), which abolished its kinase activity (the ATP-binding residue Lys-133 is required for kinase activity); a 264-site mutant (Y264F or Y264A), which blocked its autophosphorylation activity (autophosphorylation of Tyr-264 in the activation loop is required for kinase activity); and a double-site mutant (K133R/Y264F or K133A/Y264A). Reporter assays revealed that the kinase-inactive mutants of DYRK4 still enhanced SeV-induced activation of the IFNβ promoter with single or double mutations (Fig. EV2A) and promoted SeV-induced phosphorylation of IRF3 and IκBα (Fig. EV2B). Furthermore, we found that DYRK4 did not promote IFN-γ-induced IRF1 transcription (Fig. EV2C). These results indicate that the role of DYRK4 in virus-triggered signaling is independent of its kinase activity.

We further explored whether DYRK4 is involved in intracellular dsRNA-triggered signaling. By transfection into 293T and RAW264.7 cells, DYRK4 overexpression promoted the activation of the poly(I:C)-triggered IFNβ promoter (Fig. EV2D,E). The opposite phenomenon was observed with the knockdown of DYRK4 (Fig. EV2F). In addition, DYRK4 facilitated the activation of the IFNβ promoter via poly(I:C)-triggered TLR3-mediated signaling in 293-TLR3 cells and via LPS-induced TLR4-mediated signaling in RAW264.7 cells (Fig. EV2D,E). Consistently, DYRK4 knockdown inhibited the activation of the IFNβ promoter resulting from TLR3-mediated signaling triggered by poly(I:C) in 293-TLR3 cells (Fig. EV2F). These data suggest that DYRK4 is required for RLR-mediated and TLR3- and TLR4-mediated activation of IRF3 and induction of IFNβ.

## DYRK4 deficiency inhibits virus-triggered signaling

To further investigate the role of DYRK4 in type I IFN-mediated antiviral responses in vivo, we generated Dyrk4-deficient mice (Fig. EV2G). PCR, qPCR, and immunoblot analyses confirmed that Dyrk4 genomic DNA, mRNA, or protein was defective in Dyrk4$^{-/-}$ mice (Fig. EV2H–K). In addition, the number of cells in the spleen and thymus was not affected in Dyrk4$^{-/-}$ mice (Fig. EV2L). Flow cytometry analysis revealed that the percentage of myeloid cells in

the spleens of Dyrk4$^{-/-}$ mice was similar to that in the spleens of Dyrk4$^{+/+}$ mice, suggesting that Dyrk4 may not be involved in myeloid cell development (Fig. EV2M).

To further determine whether DYRK4 is necessary for virus-triggered signaling in immune cells, we infected Dyrk4$^{+/+}$ and Dyrk4$^{-/-}$ bone marrow-derived macrophages (BMDMs), dendritic cells (BMDCs) and mouse lung fibroblasts (MLFs) with SeV, VSV, and HSV-1. qPCR analysis revealed that Dyrk4 deficiency suppressed the transcription of the downstream antiviral gene Ifnb1 induced by SeV in BMDCs and MLFs (Fig. 2A) and that of VSV in BMDCs, MLFs, and BMDMs (Fig. 2B) and that of HSV-1 in MLFs (Fig. 2C). Concurrently, Dyrk4 deficiency inhibited the transcription of downstream antiviral genes induced by SeV in BMDMs (Fig. 2D), including Ifnb1, Isg15, Cxcl10, and Il6. Consistently, the phosphorylation of IRF3 and p65 induced by SeV, VSV, and HSV-1 was inhibited in Dyrk4$^{-/-}$ MLFs compared with that in wild-type MLFs (Fig. 2E,F), BMDCs (Fig. 2G) and BMDMs (Fig. 2H,I). These data suggest that Dyrk4 is required for the efficient induction of downstream antiviral genes by different types of RNA and DNA viruses in primary mouse immune cells and fibroblasts.

## Dyrk4 is essential for innate antiviral responses in vivo

To assess the importance of Dyrk4 in host defense against viral infection in vivo, we infected 8-week-old Dyrk4$^{+/+}$ and Dyrk4$^{-/-}$ mice with VSV via tail vein injections and monitored their survival rates. The results revealed that Dyrk4$^{-/-}$ mice were more sensitive to VSV-induced mortality (Fig. 3A). Similarly, compared with those in Dyrk4$^{+/+}$ mice, the mRNA expression of Ifnb1, Cxcl10, Isg15, and Il6 in the spleen, lung, and liver of Dyrk4$^{-/-}$ mice was reduced, as determined by qPCR analysis after VSV infection (Fig. 3B). ELISA revealed lower levels of IFN-β and CXCL10 in the sera of Dyrk4$^{-/-}$ mice than in the sera of Dyrk4$^{+/+}$ mice (Fig. 3C), and VSV replication was increased in the liver and lung (Fig. 3D). Hematoxylin and eosin staining revealed greater lung tissue damage in Dyrk4$^{-/-}$ mice than in Dyrk4$^{+/+}$ mice following VSV infection (Fig. 3E). These results confirm that Dyrk4 is important in the host defense against viral infection in vivo.

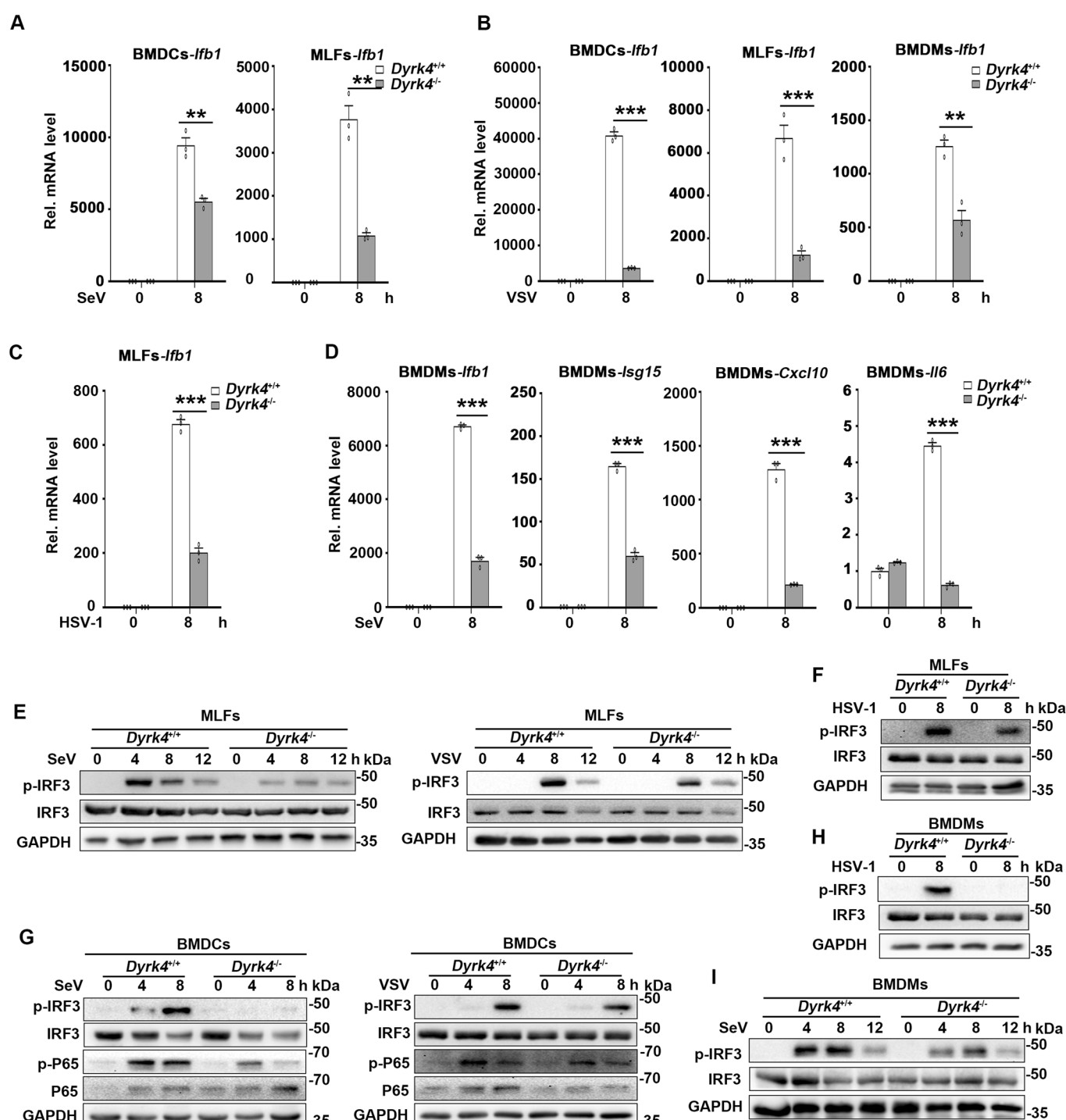

**Figure 2. Dyrk4 is required for the virus-mediated innate immune response.**

(A–C) Effects of Dyrk4 deficiency on the virus-induced transcription of *Ifnb1*. *Dyrk4*$^{+/+}$ and *Dyrk4*$^{−/−}$ BMDCs, BMDMs, and MLFs were infected with SeV, VSV or HSV-1 for 8 h before qPCR analysis. (*P* value; A; BMDCs: 0.0025, MLFs: 0.00103; B; BMDCs: 3.2 × 10$^{−6}$, MLFs: 0.00097, BMDMs: 0.0026; C; MLFs: 3.7 × 10$^{−5}$) (*n* = 3 biological replicates). (D) Effects of Dyrk4 deficiency on the virus-induced transcription of downstream antiviral genes. *Dyrk4*$^{+/+}$ and *Dyrk4*$^{−/−}$ BMDMs were infected with SeV for 8 h before qPCR analysis. (*P* value; *Ifnb1*: 3.4 × 10$^{−6}$, *Isg15*: 2.6 × 10$^{−5}$, *Cxcl10*: 3.3 × 10$^{−5}$, *Il6*: 1.6 × 10$^{−6}$) (*n* = 3 biological replicates). (E, G, I) Effects of Dyrk4 deficiency on SeV- or VSV-induced phosphorylation of IRF3 (Ser396) and p65. *Dyrk4*$^{+/+}$ and *Dyrk4*$^{−/−}$ BMDCs, MLFs, and BMDMs were infected with SeV or VSV for the indicated times before immunoblotting analysis with the indicated antibodies. (F, H) Effects of Dyrk4 deficiency on HSV-1-induced phosphorylation of IRF3 (Ser396). *Dyrk4*$^{+/+}$ and *Dyrk4*$^{−/−}$ MLFs and BMDMs were infected with HSV-1 for 8 h before immunoblotting analysis with the indicated antibodies. Data information: Data are representative of three biological replicates and are shown as the means with SEMs (A–D); data in (E–I) are representative of two replicates. **$P$ < 0.01, ***$P$ < 0.001; two-tailed unpaired Student's *t* test. Source data are available online for this figure.

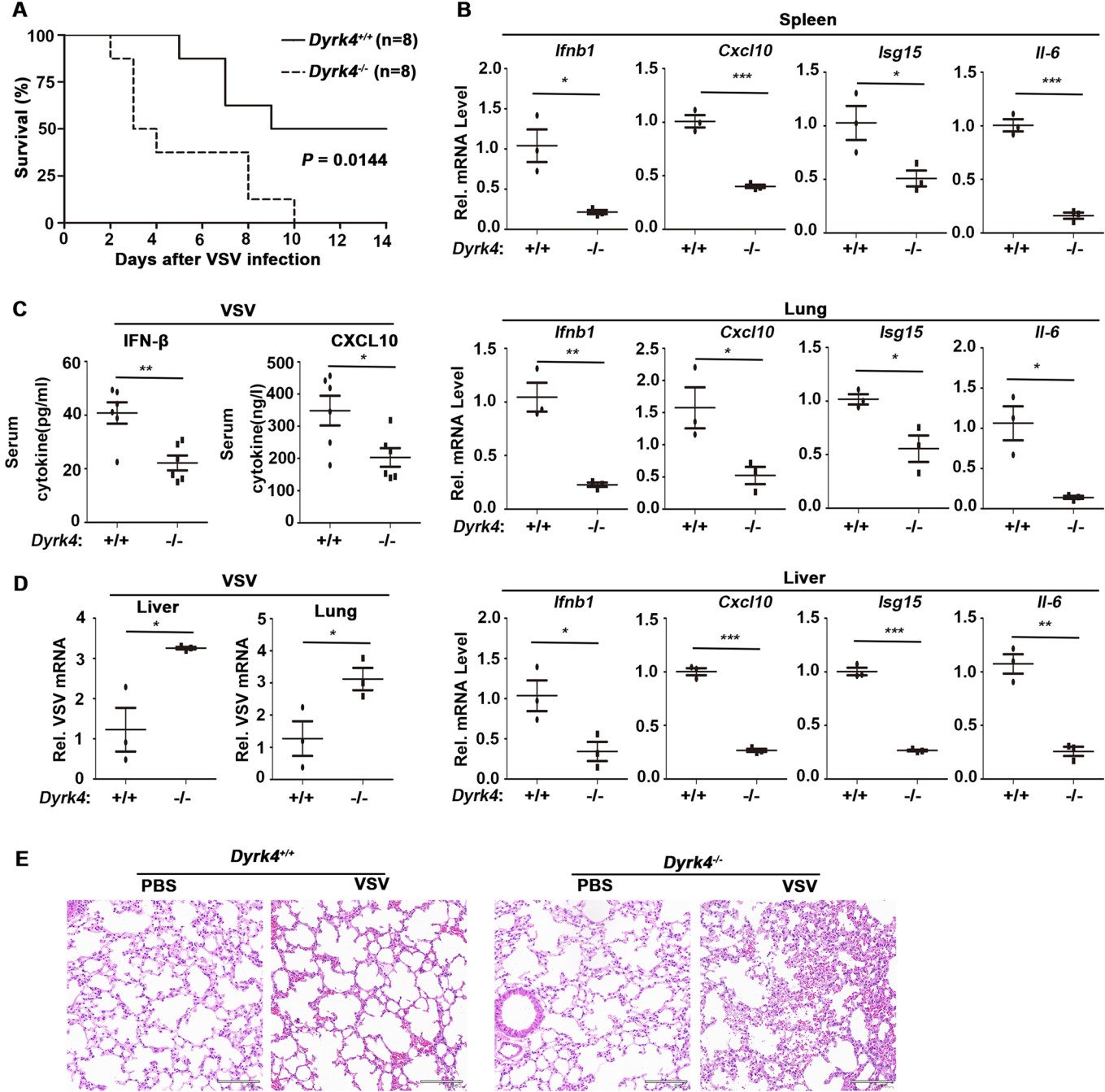

**Figure 3. Dyrk4 is essential for host defense against VSV infection in mice.**

(A) Survival (Kaplan–Meier curve) of $Dyrk4^{+/+}$ and $Dyrk4^{-/-}$ mice ($n = 8$) intravenously infected with VSV at $10^8$ PFU per mouse was monitored for 14 days. (B) qPCR analysis of $Ifnb1$, $Cxcl10$, $Isg15$, and $Il6$ mRNA in the spleen (upper), lung (middle), and liver (bottom) of $Dyrk4^{+/+}$ ($n = 3$) and $Dyrk4^{-/-}$ mice ($n = 3$) intravenously infected with VSV at $10^8$ PFU per mouse for 24 h. (P value; Spleen; $Ifnb1$: 0.0155, $Cxcl10$: 0.0005, $Isg15$: 0.0422, $Il6$: 0.0002, Lung; $Ifnb1$: 0.0038, $Cxcl10$: 0.0386, $Isg15$: 0.0256, $Il6$: 0.0121, Liver; $Ifnb1$: 0.0374, $Cxcl10$: $3.4 \times 10^{-5}$, $Isg15$: $3.3 \times 10^{-5}$, $Il6$: 0.0012). (C, D) ELISA analysis of IFN-β and CXCL10 in the sera of (C) ($n = 6$) or qPCR analysis of VSV mRNA in the liver and lung from (D) ($n = 3$) $Dyrk4^{+/+}$ and $Dyrk4^{-/-}$ mice intravenously infected with VSV at $10^8$ PFU per mouse for 16 h or 24 h, respectively. (P value; C; IFN-β: 0.0034, CXCL10: 0.0237, D; liver: 0.0205, lung: 0.0446). (E) Hematoxylin–eosin (HE) staining of lung sections from $Dyrk4^{+/+}$ and $Dyrk4^{-/-}$ mice intravenously infected with VSV as described in D. Scale bars, 100 μm. Data information: Data are representative of three biological replicates and are shown as the means with SEMs (B–D). *$P < 0.05$, **$P < 0.01$, ***$P < 0.001$; two-tailed unpaired Student's t test. Analysis of the survival curves of the mice was performed via a log-rank test. Source data are available online for this figure.

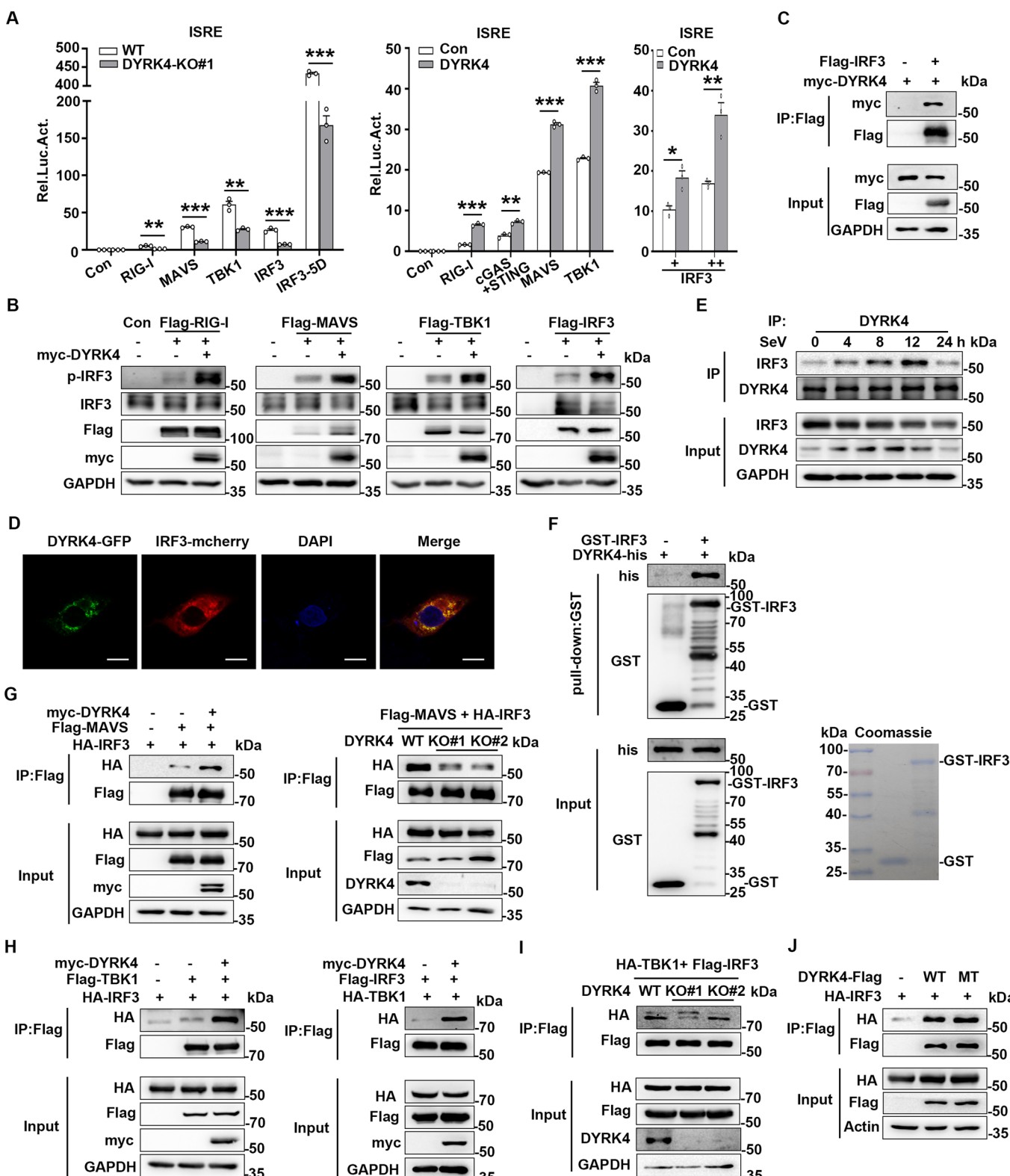

**Figure 4. DYRK4 mediates virus-triggered signaling at the level of IRF3.**

(A) Luciferase activity of ISRE in DYRK4-KO and control HEK293T cells transfected with plasmids expressing RIG-I, MAVS, TBK1, IRF3, or IRF3-5D for 24 h and in HEK293T cells transfected with plasmids expressing RIG-I, cGAS, STING, MAVS, TBK1, or IRF3 along with the DYRK4 or control vector for 24 h. (*P* value; DYRK4-KO; RIG-I: 0.0015, MAVS: $8.8 \times 10^{-6}$, TBK1: 0.0019, IRF3: 0.00026, IRF3-5D: $3.2 \times 10^{-5}$, DYRK4; RIG-I: $1.2 \times 10^{-5}$, cGAS + STING: 0.0012, MAVS: $7.3 \times 10^{-6}$, TBK1: $2.9 \times 10^{-5}$, IRF3 +: 0.0133, IRF3 + +: 0.005) ($n = 3$ biological replicates). (B) Western blot analysis of IRF3 phosphorylation in HEK293T cells co-transfected with Flag-RIG-I, Flag-MAVS, Flag-TBK1, Flag-IRF3, and myc-DYRK4 or empty vector for 24 h. (C) Coimmunoprecipitation analysis of the interaction between DYRK4 and IRF3 in HEK293T cells transfected with myc-DYRK4 and Flag-IRF3 or the control vector for 24 h. (D) Confocal microscopic analysis of HeLa cells transfected for 24 h with plasmids expressing IRF3-mCherry or DYRK4-GFP. Scale bars, 10 μm. (E) Endogenous immunoprecipitation analysis of the interaction between DYRK4 and IRF3 in HEK293T cells infected with SeV for the indicated times. (F) GST pull-down analysis of the interaction between GST-IRF3 and DYRK4-His in vitro. (G) DYRK4 promotes the association of MAVS with IRF3. HEK293T cells were co-transfected with Flag-MAVS and HA-IRF3 together with control and myc-DYRK4 plasmids (left) for 24 h; DYRK4-KO and control HEK293T cells were transfected with Flag-MAVS and HA-IRF3 plasmids (right) for 24 h before coimmunoprecipitation and immunoblotting analysis with the indicated antibodies. (H) DYRK4 promotes the interaction of TBK1 with IRF3. HEK293T cells were transfected with Flag-TBK1 and HA-IRF3 (left) or HA-TBK1 and Flag-IRF3 (right) together with a control or myc-DYRK4 plasmid for 24 h before coimmunoprecipitation and immunoblotting analysis with the indicated antibodies. (I) DYRK4-KO decreases the association of TBK1 with IRF3. DYRK4-KO and control HEK293T cells were transfected with HA-TBK1 and Flag-IRF3 plasmids for 24 h before coimmunoprecipitation and immunoblotting analysis with the indicated antibodies. (J) Coimmunoprecipitation analysis of the interaction between DYRK4-WT or -MT and IRF3 in HEK293T cells transfected with HA-IRF3 and DYRK4-Flag (WT or MT) or the control vector plasmid for 24 h. Data information: Data are representative of three biological replicates and are shown as the mean with SEM (A); data in (B–J) are representative of two replicates. *$P < 0.05$, **$P < 0.01$, ***$P < 0.001$; two-tailed unpaired Student's *t* test. Source data are available online for this figure.

## DYRK4 regulates virus-triggered signaling at the IRF3 level

To probe the mechanism of DYRK4 in the innate antiviral response, we first investigated the cellular localization of DYRK4 before and after viral infection. Although the GeneCards database revealed that the subcellular localization of DYRK4 included the nucleus, cytoskeleton, cytoplasm, endoplasmic reticulum, mitochondria, and cell membrane (plasma membrane), our cellular grading experiments revealed that DYRK4 was predominantly distributed in the cytoplasm and, to a lesser extent, in the nucleus before or after SeV infection (Fig. EV3A). Next, we sought to determine the level at which DYRK4 regulates the virus-induced IRF3 activation pathway. The results revealed that DYRK4 deficiency inhibited RIG-I, MAVS, TBK1, IRF3, and the constitutively active phosphorylation mimetic IRF3-5D-mediated ISRE activation; conversely, DYRK4 overexpression promoted RIG-I, MAVS, TBK1 and IRF3-mediated ISRE activation (Fig. 4A). In contrast, DYRK4 overexpression increased RIG-I, MAVS, TBK1, and IRF3-mediated IRF3 phosphorylation (Fig. 4B). These results indicated that DYRK4 might function at the level of IRF3. Coimmunoprecipitation experiments revealed that DYRK4 interacts with IRF3 (Fig. 4C). Confocal microscopy revealed that DYRK4 colocalized with IRF3 (Fig. 4D). We next examined whether DYRK4 was associated with IRF3 in virus-infected cells. Endogenous coimmunoprecipitation experiments revealed that DYRK4 was constitutively associated with IRF3 in uninfected cells and that the interaction between DYRK4 and IRF3 increased to high levels at 8 and 12 h of viral infection (Fig. 4E). Furthermore, in vitro pull-down analysis confirmed that recombinant DYRK4 binds to IRF3 (Fig. 4F). These results suggest that DYRK4 is associated with IRF3 in a virus infection-dependent manner.

DYRK4 contains an N-terminal, kinase structural (K), and C-terminal domain (Aranda et al, 2011). In contrast, IRF3 contains a DBD, a TAD transcriptional activation domain, and an RD nuclear response domain (Wu et al, 2020), and different domain mutants of DYRK4 or IRF3 have been constructed (Fig. EV3B,C). Domain mapping experiments revealed that the K kinase domain of DYRK4 interacts with the IAD domain of IRF3 (Fig. EV3B,C).

Reporter assays revealed that the K domain of DYRK4, which can interact with IRF3, also enhances SeV-induced activation of the IFN-β promoter. In contrast, the C-terminus of DYRK4, which is unable to interact with IRF3, inhibits SeV-induced activation of the IFN-β promoter. Moreover, the N-terminus alone had a weak effect on SeV-induced activation of the IFN-β promoter (Fig. EV3D). Consistently, reconstitution of the K domain in DYRK4-KO cells restored SeV-induced activation of ISRE (Fig. EV3E), and these results suggest that the binding of DYRK4 and IRF3 is required for DYRK4 function in IRF3-mediated signaling.

Interestingly, coimmunoprecipitation experiments revealed that DYRK4 overexpression increased the association of IRF3 with MAVS (Fig. 4G, left panel) and TBK1 (Fig. 4H). In contrast, DYRK4 deficiency in cells had the opposite effect (Fig. 4G, right panel and I). Moreover, the overexpression of DYRK4-WT and DYRK4-MT (K133R/Y264F) kinase inactivating mutants resulted in similar binding capacities for IRF3 (Fig. 4J). These results confirmed that DYRK4 directly interacts with IRF3 and promotes the formation of the IRF3 signaling complex.

## DYRK4 promotes the polyubiquitination of IRF3

Since DYRK4 affects IRF3 function independent of its kinase activity, we speculate that DYRK4 affects IRF3 function in other ways, as DYRK2, a member of the DYRK family, recruits NLRP4 and the E3 ubiquitin ligase DTX4 to degrade TBK1 (An et al, 2015). It has also been reported that DYRK2 is a scaffold that promotes the assembly of E3 ligases (Maddika and Chen, 2009). Wu et al confirmed that viral infection increases the ubiquitination level of IRF3 and the degradation of IRF3 (Wu et al, 2020). We first tested whether DYRK4 affects the stability of the IRF3 protein by recruiting E3 ubiquitin ligases to ubiquitinate IRF3. The results revealed that overexpression of DYRK4 stabilized the exogenous or endogenous IRF3 protein (Fig. 5A,B), and under conditions of viral infection, overexpression of DYRK4 extended the half-life of the exogenous IRF3 protein (Fig. 5C) but did not affect IRF3 transcription (Fig. 5D). In addition, our results revealed that overexpression of DYRK4-MT (a kinase-inactive mutant) promoted the stability of the endogenous IRF3 protein (Fig. 5B, right panel). Furthermore, coimmunoprecipitation experiments revealed

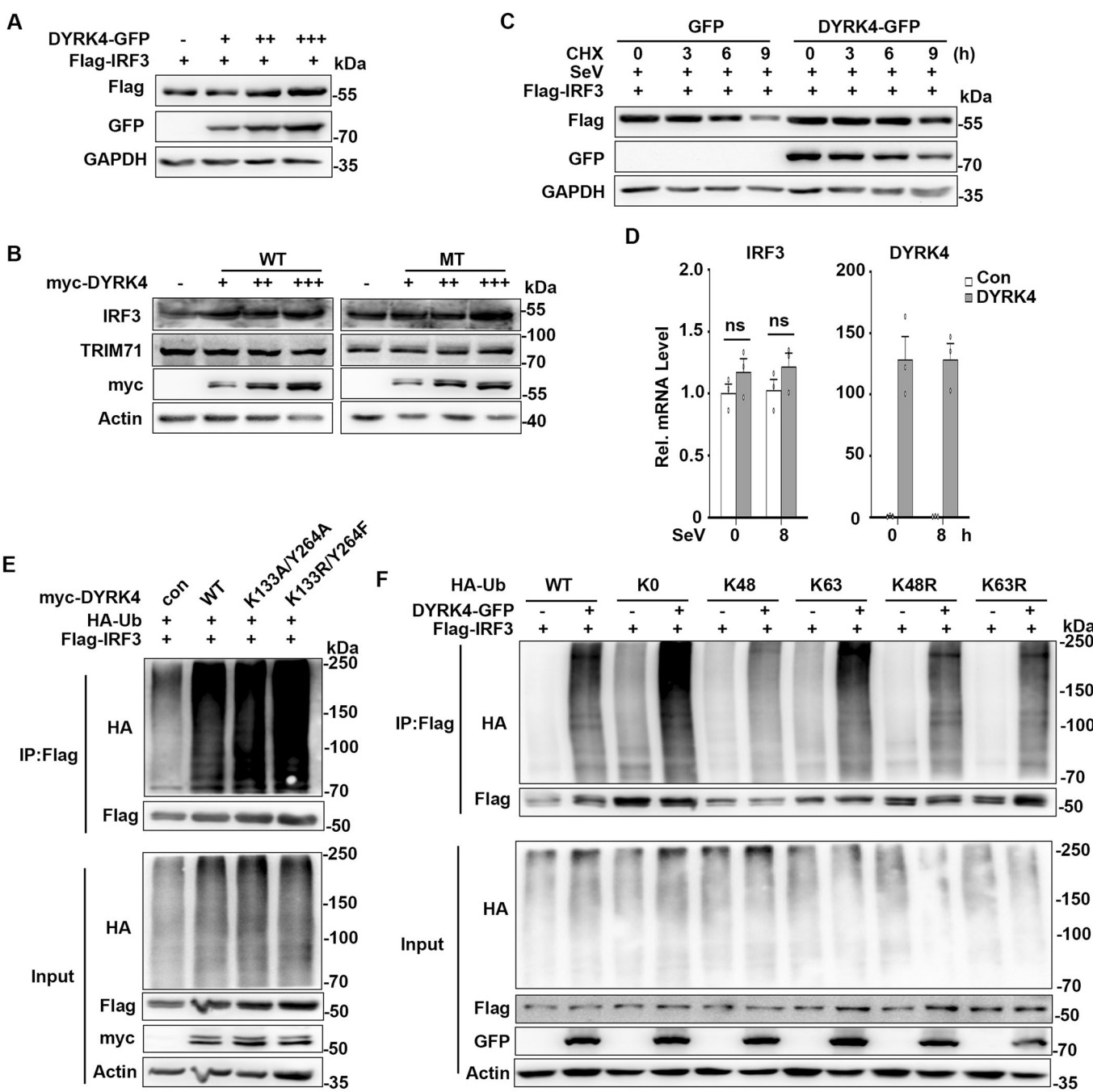

**Figure 5. DYRK4 enhances the polyubiquitination of IRF3.**

(A) Immunoblot analysis of the protein level of IRF3 in HEK293T cells transfected with Flag-IRF3 in fixed amounts and with different concentrations of DYRK4 for 24 h. (B) Immunoblot analysis of the protein level of endogenous IRF3 in HEK293T cells transfected with the DYRK4-WT or MT plasmid at different dosages for 24 h. (C) Immunoblot analysis of the protein level of IRF3 in HEK293T cells transfected with Flag-IRF3 and DYRK4 for 20 h, preinfected with SeV for 12 h, and then treated with CHX for the indicated times. (D) qPCR analysis of the mRNA level of endogenous IRF3 in HEK293T cells transfected with the DYRK4 plasmid for 24 h and then infected with SeV for 8 h. (ns: not significant) ($n = 3$ biological replicates). (E) Effects of DYRK4 and its mutants on the polyubiquitination of IRF3. HEK293T cells were transfected with Flag-IRF3 and HA-Ub together with a control or DYRK4 and its mutant plasmids for 24 h, followed by immunoblotting and coimmunoprecipitation analysis with the indicated antibodies. (F) Effects of DYRK4 on the polyubiquitination of IRF3. HEK293T cells were transfected with Flag-IRF3 and HA-Ub or its mutants together with a control or DYRK4 plasmid for 24 h, followed by immunoblotting and coimmunoprecipitation analysis with the indicated antibodies. Data information: Data are representative of three biological replicates and are shown as the mean with SEM (D); data in (A–C, E, F) are representative of two replicates. ns not significant; two-tailed unpaired Student's $t$ test. Source data are available online for this figure.

that overexpression of DYRK4 and DYRK4-MT caused increased ubiquitination of IRF3 (Fig. 5E). These results suggest that DYRK4 stabilizes IRF3 protein levels and promotes the poly-ubiquitination of IRF3 and that this process is not dependent on its kinase activity. Since the interaction between DYRK4 and IRF3 depends on viral infection (Fig. 4E), DYRK4 affects IRF3 ubiquitination during viral infection. SeV infection upregulated IRF3 ubiquitination in control HEK293T cells. In contrast, DYRK4 deficiency reduced the virus-induced ubiquitination of IRF3 (Fig. EV3F). These results indicate that DYRK4 targets the ubiquitination of IRF3 during viral infection. Since different ubiquitin chain linkages have different functions, we next investigated which polyubiquitination mechanism is mediated by DYRK4. The overexpression of DYRK4 promoted not only K63-linked and K48-linked but also K63R-linked and K48R-linked polyubiquitination of IRF3. Interestingly, we found that over-expression of DYRK4 promoted multiple ubiquitin chains (Fig. 5F), suggesting that DYRK4 may affect the K0 (a lysine-free Ub mutant) linear ubiquitination of IRF3.

## DYRK4 mediates the interaction of IRF3 with TRIM71

Since DYRK4 is not an E3 ubiquitin ligase, we speculate that it functions as a scaffold protein that recruits E3 ubiquitin ligases to affect the ubiquitination of IRF3. To identify the E3 ligase that interacts with DYRK4, we performed mass spectrometry (MS) analysis of DYRK4 and focused on several E3 ubiquitin ligases, such as TRIM4, TRIM71, TRIM21, RNF138, and TRIM23 (Dataset EV1). Reporter assays indicated that TRIM71 promoted IRF3-mediated ISRE activation (Fig. 6A). Furthermore, TRIM71 significantly promoted SeV-induced ISRE activation (Fig. 6B). In contrast, the knockdown of TRIM71 inhibited SeV-induced ISRE activation (Fig. EV3G). To determine how DYRK4 affects the activation of IRF3-mediated ISRE via TRIM71, we first showed that DYRK4 interacted with TRIM71 via co-IP (Fig. 6C). Domain mapping experiments revealed that the N-terminal and K kinase domains of DYRK4 with the RING domain of TRIM71 were required for their interaction (Fig. EV3H,J). More importantly, IRF3 also bound to TRIM71 (Fig. 6D). Domain mapping experiments revealed that the IAD domain of IRF3 and the RING domain of TRIM71 were required for their interaction (Fig. EV3I,K). Coimmunoprecipitation experiments revealed that overexpression of both DYRK4 and the DYRK4-MT mutant promoted the interaction of TRIM71 with IRF3 (Fig. 6E,F). However, the knockout of DYRK4 reduced the interaction between TRIM71 and IRF3 (Fig. 6G). Consistently, DYRK4 synergized with TRIM71 and IRF3 to activate ISRE in a dose-dependent manner (Fig. 6H), whereas DYRK4 deficiency inhibited SeV-induced ISRE activation by TRIM71 (Fig. 6I). These results suggest that DYRK4 can act as a scaffold protein to enhance the IRF3–TRIM71 interaction, hence promoting antiviral immunity, which is independent of its enzymatic activity.

## DYRK4 acts through TRIM71

To determine the mechanisms by which TRIM71 functions in DYRK4-mediated regulation of IRF3 activation, we first investigated whether TRIM71 mediates the polyubiquitination of IRF3. Ubiquitination analysis revealed that TRIM71 promoted the polyubiquitination of IRF3 (Fig. EV4A). However, the ability of TRIM71-CA (enzyme inactivation mutant C12A/C15A, which lies in the RING domain) and ΔRING (RING domain deletion mutant) to increase the ubiquitination of IRF3 was lost (Fig. 7A). Moreover, the ability of TRIM71-CA to promote SeV-induced and IRF3-mediated ISRE activation was attenuated compared with that of TRIM71-WT (Fig. 7B). These results suggest that TRIM71 regulates IRF3 in an enzymatic activity-dependent manner. We next determined the type of ubiquitination mediated by TRIM71. By cotransfecting IRF3 with ubiquitin mutants containing only one lysine residue (K) or with only one lysine residue mutated to arginine (KR) and with ubiquitin mutants containing no lysine residues (K0), we found that TRIM71 promoted the formation of multiple ubiquitin chains (Fig. EV5A,B). These results suggest that TRIM71 mediates the K0-linked linear ubiquitination of IRF3. Furthermore, coimmunoprecipitation experiments revealed that DYRK4 overexpression did not affect the polyubiquitination of IRF3 in TRIM71-deficient cells (Fig. 7C). Notably, TRIM71 deficiency inhibited DYRK4-mediated ISRE activation during SeV infection (Fig. 7D). These results suggest that DYRK4 mediates the polyubiquitination and activation of IRF3 mainly through TRIM71.

We next sought to determine the biological importance of the linear ubiquitination of IRF3. Compared with control cells, SeV-induced transcription of IFNB1 and CXCL10 was inhibited in TRIM71-deficient HEK293T cells (Fig. EV4B), in which SeV-induced phosphorylation of IRF3, p65, and IκBα (Fig. EV4C), IRF3 dimerization (Fig. EV4D), and nuclear translocation of IRF3 (Fig. EV4E) were also inhibited. We also determined whether TRIM71 physically interacted with IRF3 in untransfected cells. Endogenous immunoprecipitation experiments revealed that TRIM71 interacts with IRF3 under physiological conditions (Fig. EV4F), however, their interaction was weak under resting conditions and reached its highest level 8 h after SeV infection (Fig. EV4G). These results suggest that the association of TRIM71 with IRF3 increases in a viral infection-dependent manner. Furthermore, we observed that TRIM71 knockout attenuated SeV-induced ubiquitination of endogenous (Fig. 7E) and K0-linked exogenous IRF3 (Fig. 7F). Overall, our findings suggest that IRF3 undergoes TRIM71-induced linear ubiquitination. To explore which lysine residue TRIM71 ubiquitinates IRF3, we mutated fourteen ubiquitinated lysine (K) residues to arginine (R) residues on IRF3 and found that mutations in K5, K87, K105, K313, or K366 significantly reduced the ubiquitination of IRF3 by TRIM71 compared with that of wild-type (WT) IRF3 (Fig. EV5C,D). Furthermore, reporter assays revealed that only mutations in K87 or K105 significantly reduced SeV-induced activation of ISRE (Fig. EV5E). In addition, the double mutation of IRF3 in K87 and K105 (K87/105 R) also significantly reduced the ubiquitination of IRF3 by TRIM71 (Fig. 7G), whereas the mutation of either K87 or K105 or the double mutation (K87/105R) attenuated TRIM71-promoted IRF3-mediated ISRE activation (Fig. 7H). Importantly, K87 and K105 were highly conserved in *Homo sapiens*, *Caenorhabditis elegans*, horses, brown house mice, and African clawed toads (Fig. 7I). We then investigated whether TRIM71 affects IRF3 protein stability and found that the overexpression of TRIM71 to a certain dose stabilized exogenous IRF3 protein levels (Fig. EV4H), with little to no effect on IRF3 mRNA levels (Fig. EV4I). IRF3 has been shown to undergo autophagic degradation in response to viral infection(Wu et al, 2020). In 293T cells, we replicated this result

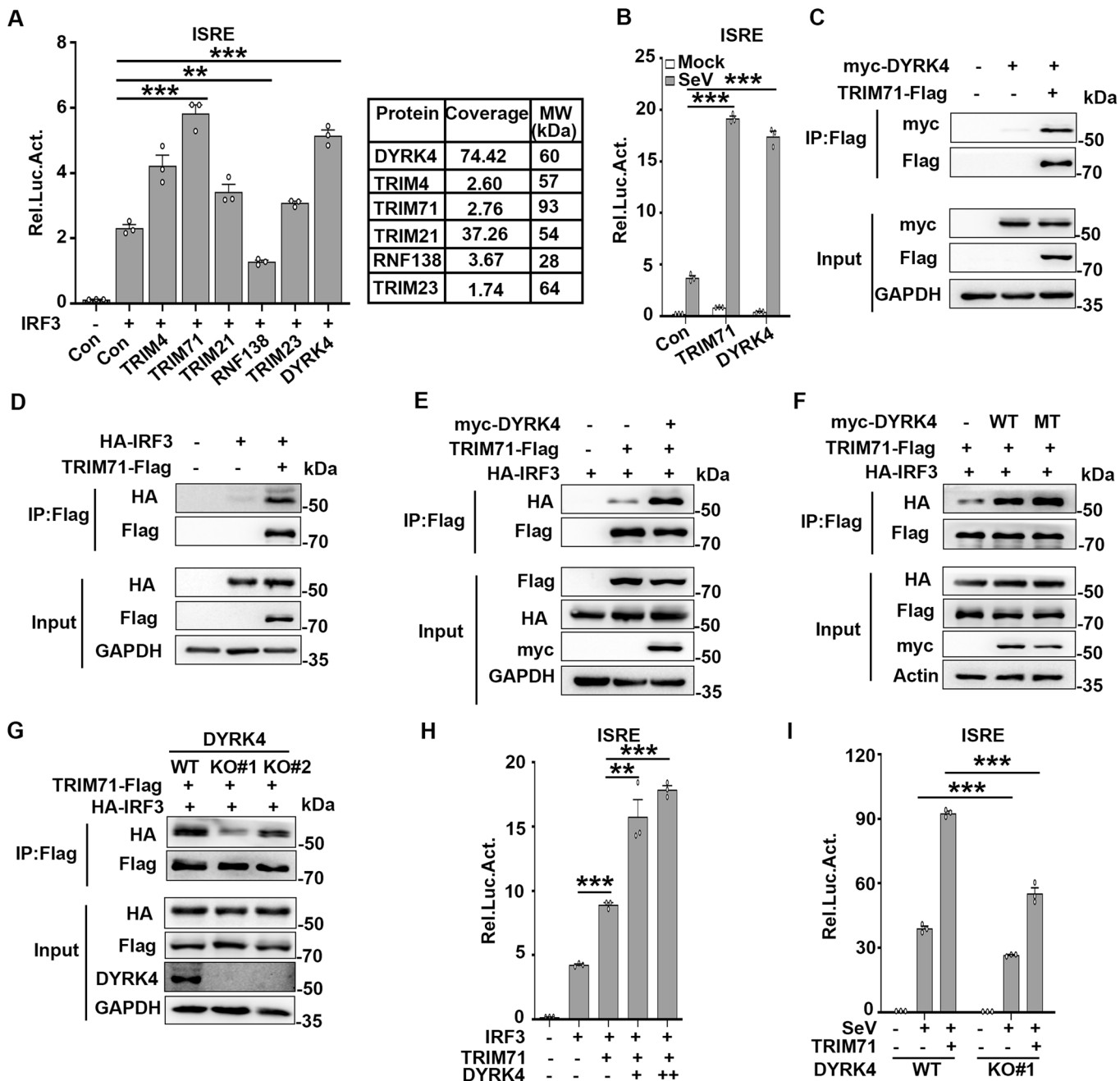

(Fig. EV4J), and we found that overexpression of TRIM71 promoted the stabilization of endogenous IRF3 upon SeV infection (Fig. 7J). However, K87 or K105 mutations or double mutations (K87/105R) attenuated the effect of TRIM71 on IRF3 stabilization (Fig. 7K). More importantly, the double mutant IRF3 degraded rapidly upon viral infection (Fig. EV4K). In addition, overexpression of TRIM71 similarly promoted the stability of IRF3-2A and IRF3-5D (Fig. EV4L). TRIM71 knockout during SeV infection increased IRF3 degradation (Fig. 7L). Moreover, reconstitution of TRIM71-WT but not TRIM71-CA in TRIM71-KO cells promoted IRF3 stability (Fig. 7M), and reconstitution of TRIM71 in TRIM7-KO cells enhanced SeV-induced activation of the ISRE

(Fig. 7N), suggesting that TRIM71-mediated linear ubiquitination of IRF3 promotes its stability and activation. Overall, these results suggest that DYRK4 mediates linear ubiquitination and activation of IRF3 at K87/105 by recruiting TRIM71, which promotes the stability of IRF3 and, thus, the antiviral response.

## DYRK4 and TRIM71 interact with LUBAC

The linear Ub chain assembly complex (LUBAC) is the only E3 ubiquitin ligase reported thus far that can catalyze the generation of linear ubiquitin ligases. It consists of HOIL-1L, HOIP, and SHARPIN, a Ub ligase complex that specifically induces linear

**Figure 6. DYRK4 mediates the interaction of TRIM71 with IRF3.**

(A) Effects of Flag-tagged E3s and DYRK4 on IRF3-mediated activation of the ISRE. HEK293T cells were transfected with the ISRE reporter and HA-IRF3 with the control or the indicated plasmids for 24 h before luciferase assays (left). a Mass spectrometry (MS) analysis of DYRK4 (right). (P value; TRIM71: 0.00026, RNF138: 0.00106, DYRK4: 0.00015) ($n = 3$ biological replicates). (B) TRIM71 and DYRK4 activate ISRE. HEK293T cells were transfected with the ISRE reporter and a control, TRIM71, DYRK4 plasmids for 24 h and then infected with SeV for 12 h before luciferase assays. (P value; TRIM71: $9.8 \times 10^{-7}$, DYRK4: $1.5 \times 10^{-5}$) ($n = 3$ biological replicates). (C) Coimmunoprecipitation analysis of the interaction between DYRK4 and TRIM71 in HEK293T cells transfected with myc-DYRK4 and TRIM71-Flag or the control plasmid for 24 h. (D) Coimmunoprecipitation analysis of the interaction between IRF3 and TRIM71 in HEK293T cells transfected with HA-IRF3 and the TRIM71-Flag or control plasmid for 24 h. (E) DYRK4 promotes the interaction of TRIM71 with IRF3. HEK293T cells were transfected with TRIM71-Flag or HA-IRF3 and control or myc-DYRK4 plasmids for 24 h before immunoblotting and coimmunoprecipitation analysis with the indicated antibodies. (F) DYRK4-WT or -MT promotes the interaction of TRIM71 with IRF3. The experiments were performed similarly to those in (E). (G) DYRK4 deficiency decreases the association of TRIM71 with IRF3. DYRK4-KO and control HEK293T cells were transfected with TRIM71-Flag and HA-IRF3 for 24 h before immunoblotting and coimmunoprecipitation analysis with the indicated antibodies. (H) DYRK4 collaborates with TRIM71 to promote IRF3-mediated activation of the ISRE. HEK293T cells were transfected with the ISRE reporter and HA-IRF3 or TRIM71-Flag and the control or myc-DYRK4 plasmid for 24 h before luciferase assays. (P value; TRIM71 + : $1.3 \times 10^{-5}$, TRIM71 + DYRK4 + : 0.00716, TRIM71 + DYRK4 + + : $1.5 \times 10^{-5}$) ($n = 3$ biological replicates). (I) DYRK4 deficiency inhibits TRIM71- and SeV-mediated activation of the ISRE. DYRK4-KO and control HEK293T cells were transfected with the ISRE reporter and TRIM71 for 24 h and then infected with SeV for 12 h before luciferase assays. (P value; DYRK4-KO + TRIM71-: 0.00039, DYRK4-KO + TRIM71 + : 0.00018) ($n = 3$ biological replicates). Data information: Data are representative of three biological replicates and are shown as the means with SEMs (A, B, H, I); data in (C–G) are representative of two replicates. **$P < 0.01$, ***$P < 0.001$; two-tailed unpaired Student's $t$ test. Source data are available online for this figure.

polyubiquitination of protein substrates (Tokunaga and Iwai, 2012; Kirisako et al, 2006; Belgnaoui et al, 2012). Next, we explored whether DYRK4-mediated K0 linear ubiquitination of IRF3 through TRIM71 is associated with the LUBAC complex. Coimmunoprecipitation experiments revealed that TRIM71 can interact with the LUBAC complex (Fig. 8A) and that DYRK4 can also interact with the LUBAC complex (Fig. 8B). These results preliminarily show that DYRK4 mediates K0 linear ubiquitination of IRF3 through TRIM71 and is associated with the LUBAC complex, indicating that DYRK4, TRIM71, and LUBAC may form a large complex to mediate K0 linear ubiquitination of IRF3 synergistically.

## Discussion

Virus-triggered type I interferon induction is temporally and spatially regulated by various molecules and mechanisms (Honda et al, 2006; Yoneyama and Fujita, 2009). Although IRF3 clearly plays a crucial role in the antiviral response, the mechanisms regulating its stability and activity are not known. Linear ubiquitination in posttranslational modifications (PTMs) is a unique type of ubiquitination that opens a new door to investigating the relationship between protein ubiquitination and signaling regulation (Kirisako et al, 2006). In this study, we found that DYRK4 is essential for virus-triggered activation of IRF3 and NF-κB, IFNβ induction, and cellular antiviral responses and that this process is independent of its kinase activity; instead, DYRK4 acts as a scaffold protein to recruit TRIM71 to mediate linear ubiquitination of IRF3 and thus regulate IRF3 stability and activation. Knockout in mice revealed that DYRK4 deficiency suppressed virus-induced transcription of downstream antiviral genes in various fibroblasts and immune cells and rendered mice more susceptible to VSV-induced death. This study establishes important roles for DYRK4 in virus-triggered IRF3 activation, IFNβ induction, and the cellular antiviral response and provides a mechanistic explanation of how IRF3 remains stable before and after viral infection.

Through mass spectrometry (MS) experiments, we identified the E3 ubiquitin ligase TRIM71. TRIM71 is involved in embryonic development, neurogenesis, and stem cell renewal by degrading its bound mRNA or inhibiting its translation through various mechanisms (Li et al, 2019; Worringer et al, 2014; Aeschimann et al, 2017; Rybak et al, 2009). Recently, TRIM71 was reported to inhibit ovarian tumorigenesis by degrading mutant p53 (Chen et al, 2019). Researchers have rarely reported whether and how TRIM71 regulates SeV-triggered IRF3 activation and IFN-β induction in antiviral natural immunity. In the present study, we found that TRIM71 interacted with IRF3 in the resting state, with the interaction reaching a maximum at 8 h after SeV infection. TRIM71 enhanced SeV-induced ISRE activation, while TRIM71 knockdown had the opposite effect, and TRIM71-CA enzyme inactivation mutants attenuated the promotion of SeV-induced or IRF3-mediated ISRE activation compared with TRIM71-WT. TRIM71 knockout inhibited the phosphorylation, dimerization, and nuclear translocation of IRF3. We found that TRIM71 ubiquitinates IRF3 and that TRIM71 knockout inhibits SeV-induced ubiquitination of IRF3. Furthermore, TRIM71 modifies linear ubiquitination at K87/105 of IRF3, but detailed studies of the K87/105 site are still needed. Finally, we found that during viral infection, the overexpression of TRIM71 stabilized IRF3, and TRIM71 knockout inhibited the stabilization of IRF3. Hence, TRIM71 promoted SeV-triggered IRF3 activation and IFN-β induction through linear ubiquitination and stabilization of IRF3. These findings establish a critical role for TRIM71 in the antiviral innate immune response. However, the mechanism by which TRIM71 is recruited by DYRK4 and its detailed mechanism of IRF3 activation still need further study.

In RNA virus-infected cells, the transcriptional activation of IRF3 is triggered primarily by the RIG-I-like receptor (RLR), which also activates the RLR-induced IRF3-mediated apoptosis pathway (RIPA). The validation of RIPA as another branch of the IRF3-mediated host antiviral innate immune response has been reported in the literature. Unlike the activation of IRF3 as a transcription factor by phosphorylation only, in RIPA, the activation of IRF3 is not achieved by K63-linked polyubiquitination but by linear ubiquitination of specific Lys residues K193 and K313 (Chattopadhyay et al, 2016). Although the same linear ubiquitination of IRF3 is needed, our results differ from this finding. Firstly, the activation of IRF3 by TRIM71 is achieved by linear ubiquitination of specific Lys residues K87 and K105, unlike K193 and K313, which are used in the antiviral activation of IRF3 by RIPA;

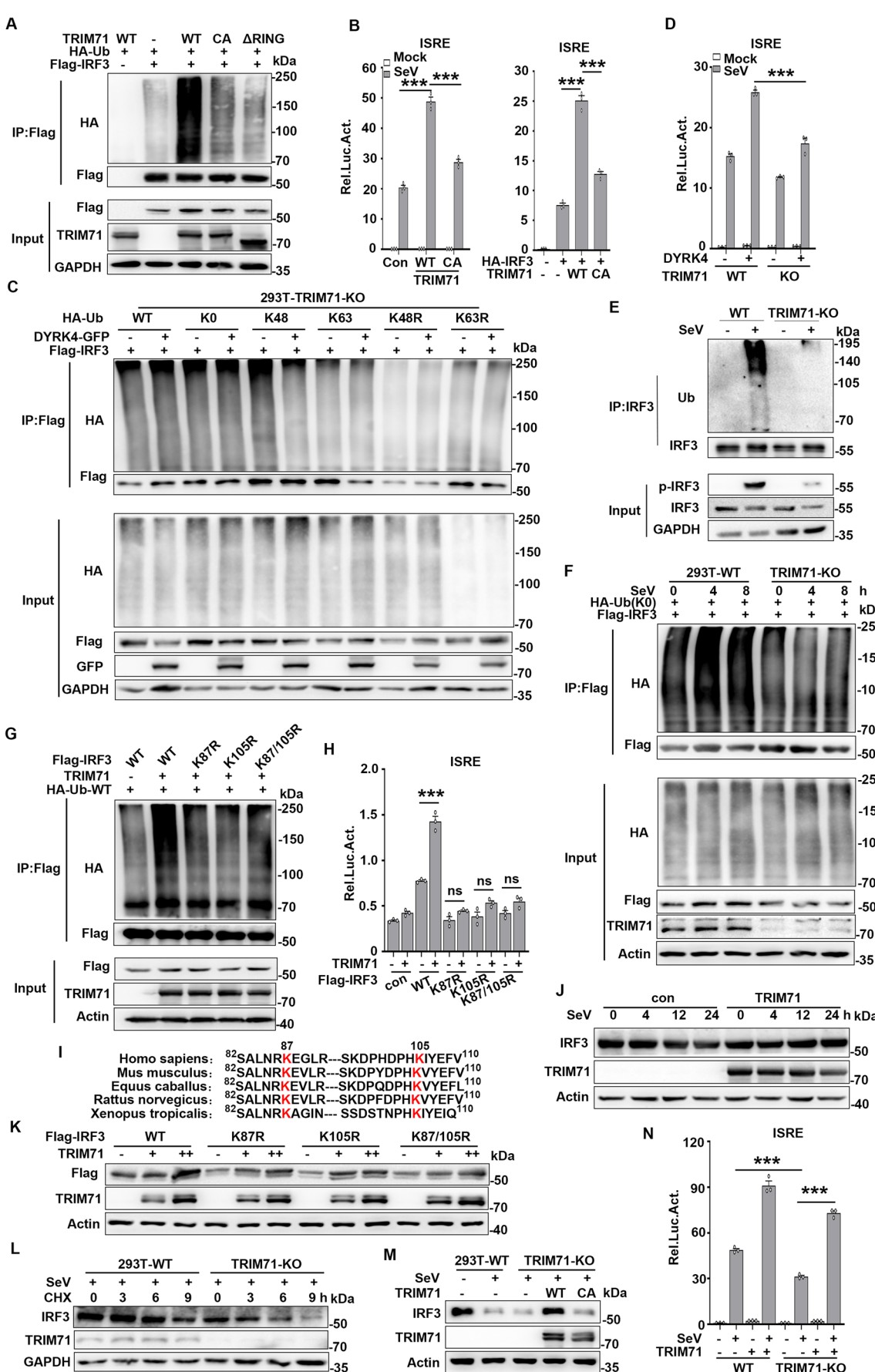

### Figure 7. DYRK4 acts through TRIM71, which promotes IRF3 stability and activation.

(A) Effects of TRIM71 and its mutants on the polyubiquitination of IRF3. HEK293T cells were transfected with Flag-IRF3 and HA-Ub together with the control, TRIM71-WT, TRIM71-CA, and TRIM71-ΔRING plasmids for 24 h, followed by immunoblotting and coimmunoprecipitation analysis with the indicated antibodies. (B) Effects of TRIM71-WT and TRIM71-CA on SeV-induced or IRF3-mediated activation of the ISRE. HEK293T cells were transfected with the ISRE reporter and the control, TRIM71-WT, or TRIM71-CA plasmid for 24 h and then infected with SeV for 12 h (left) or transfected with the ISRE reporter and HA-IRF3 together with the control, TRIM71-WT, or TRIM71-CA plasmid for 24 h (right panel) before luciferase assays. (P value; TRIM71-WT: $7.1 \times 10^{-5}$, TRIM71-CA: 0.00034, TRIM71-WT + IRF3: $4.0 \times 10^{-5}$, TRIM71-CA + IRF3: 0.00019) ($n = 3$ biological replicates). (C) Effects of TRIM71 deficiency on DYRK4-mediated polyubiquitination of IRF3. TRIM71-KO cells were transfected with Flag-IRF3 and HA-Ub or its mutants together with a control or DYRK4 plasmid for 24 h, followed by immunoblotting and coimmunoprecipitation analysis with the indicated antibodies. (D) Effects of TRIM71 deficiency on DYRK4-mediated activation of ISRE. TRIM71-KO and control HEK293T cells were transfected with DYRK4-Flag for 24 h and then infected with SeV for 12 h before luciferase assays (P value = 0.00084) ($n = 3$ biological replicates). (E) Effects of TRIM71 deficiency on SeV-induced polyubiquitination of IRF3. TRIM71-KO and control HEK293T cells were infected with SeV for 8 h before immunoblotting and immunoprecipitation analysis with the indicated antibodies. (F) Effects of TRIM71 deficiency on SeV-induced K0-linked polyubiquitination of IRF3. TRIM71-KO and control HEK293T cells were transfected with Flag-IRF3 and HA-Ub (K0) for 24 h and then infected with SeV for the indicated times before immunoblotting and coimmunoprecipitation analysis with the indicated antibodies. (G) Effects of TRIM71 on the polyubiquitination of IRF3 and its mutants. HEK293T cells were transfected with Flag-IRF3 and its mutants and HA-Ub with a control or TRIM71 plasmid for 24 h, followed by immunoblotting and coimmunoprecipitation analysis with the indicated antibodies. (H) Effects of TRIM71 on IRF3 and its mutants mediated of ISRE. HEK293T cells were transfected with the ISRE reporter and Flag-IRF3 and its mutants with control or TRIM71 plasmids for 24 h before luciferase assays. (P value; WT: 0.00037, ns: not significant) ($n = 3$ biological replicates). (I) Highly conserved lysine (K) residues (K87 and K105) on IRF3 from different species. (J) Immunoblot analysis of the protein level of IRF3 in HEK293T cells transfected with control or TRIM71 for 24 h and then infected with SeV for the indicated times. (K) Immunoblot analysis of the protein level of IRF3 in HEK293T cells transfected with Flag-IRF3 and its mutants together with TRIM71 at different dosages for 24 h. (L) TRIM71-KO and control HEK293T cells preinfected with SeV for 12 h were treated with CHX for the indicated times. The protein level of IRF3 was detected by immunoblotting analysis. (M) Immunoblot analysis of the protein level of IRF3 in TRIM71-KO and control HEK293T cells transfected with a control or the TRIM71-WT or CA mutant for 18 h and then infected with SeV for 24 h. (N) Effects of TRIM71 KO on SeV-induced activation of ISRE. TRIM71-KO and control HEK293T cells were transfected with TRIM71 plasmids for 24 h and then infected with SeV for 12 h before luciferase assays. (P value; TRIM71-KO+SeV: 0.00013, TRIM71-KO+SeV+TRIM71: $1.7 \times 10^{-5}$) ($n = 3$ biological replicates). Data information: Data are representative of three biological replicates and are shown as the means with SEMs (B, D, H, N); data in (A, C, E–G, J–M) are representative of two replicates. *$P < 0.05$, ***$P < 0.001$, ns not significant; two-tailed unpaired Student's t test. Source data are available online for this figure.

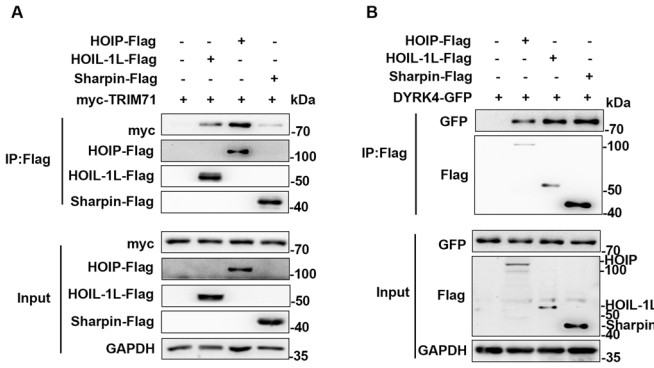

### Figure 8. TRIM71 and DYRK4 interact with LUBAC.

(A) Coimmunoprecipitation analysis of the interaction between TRIM71 and the LUBAC complex in HEK293T cells transfected with myc-TRIM71 and LUBAC complex members (HOIP-Flag, HOIL-1L-Flag and Sharpin-Flag) or control plasmids for 24 h. (B) Coimmunoprecipitation analysis of the interaction between DYRK4 and the LUBAC complex in HEK293T cells transfected with DYRK4-GFP and LUBAC complex members (HOIP-Flag, HOIL-1L-Flag and Sharpin-Flag) or control plasmids for 24 h. Data information: Data are representative of two replicates. Source data are available online for this figure.

secondly, K87R and K105R of IRF3 do not affect the antiviral effect of RIPA-activated IRF3 but inhibit its transcriptional activation. Linear ubiquitination of IRF3 is carried out by the LUBAC enzyme complex, which contains the SHARPIN, HOIP, and HOIL-1L proteins (Chattopadhyay et al, 2016). Consistently, coimmunoprecipitation experiments revealed that TRIM71 can interact with the LUBAC complex, which DYRK4 can also interact with. These results preliminarily show that DYRK4, TRIM71, and LUBAC may form a large complex to mediate the K0 linear ubiquitination of IRF3 cooperatively. However, the mechanism by which TRIM71

and DYRK4 ubiquitinate IRF3 and the detailed mechanism of IRF3 activation still need further study.

Furthermore, our results explain how cells maintain IRF3 stability under uninfected conditions, as the constitutive binding of TRIM71 to IRF3 suggests that IRF3 may maintain itself in the resting state through weak linear ubiquitination modifications and that after viral infection, DYRK4 recruits more TRIM71 and LUBAC to enhance interactions with IRF3 for increased linear ubiquitination to maintain IRF3 stability and enhance the antiviral response. However, further validation of this phenomenon with specific linear ubiquitination antibodies is needed. It has been reported that the E3 ubiquitin ligase RAUL can consistently and efficiently degrade IRF3 under physiological conditions (Yu and Hayward, 2010), so there should be a signal or modification to protect IRF3 from degradation. In contrast, the constitutive binding of TRIM71 to IRF3 suggests that IRF3 may be stabilized by linear ubiquitination to antagonize the degradation of ubiquitinated K48 linkages.

Based on our findings, we propose the following hypothesis to explain how DYRK4 positively regulates IRF3-mediated antiviral responses. In uninfected cells, DYRK4 interacts weakly with TRIM71 and IRF3. Once cells are infected, the mRNA and protein levels of DYRK4 are increased, and DYRK4 acts as a scaffold protein to recruit more TRIM71 and LUBAC to IRF3, increasing IRF3 linear ubiquitination, maintaining IRF3 stability and activation during viral infection, and promoting the IRF3-mediated antiviral response (Fig. 9). Although the exact mechanism by which DYRK4, TRIM71, and LUBAC enzyme complexes synergistically mediate IRF3 activation requires further study, our studies suggest that DYRK4 regulates the virus-triggered IRF3 and NF-κB activation pathways. Establishing a regulatory role for DYRK4 in the viral-triggered IRF3 and NF-kB activation pathways has helped elucidate the complex molecular mechanisms underlying the cellular antiviral response.

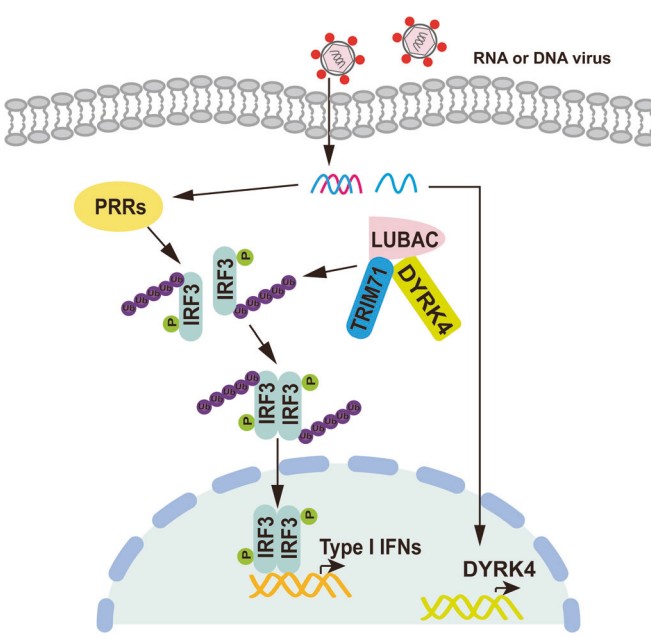

**Figure 9. Working model of DYRK4 in the regulation of the antiviral innate immune response.**

Once cells are infected, the mRNA and protein levels of DYRK4 are increased, and DYRK4 acts as a scaffold protein to recruit TRIM71 and LUBAC to IRF3, increasing IRF3 linear ubiquitination, maintaining IRF3 stability and activation during viral infection, and promoting the IRF3-mediated antiviral response. Source data are available online for this figure.

# Methods

### Reagents and tools table

| Reagent/resource | Reference or source | Identifier or catalog number |
|---|---|---|
| **Experimental models** | | |
| C57BL/6-*Dyrk4*−/− mice | Cyagen Biosciences Inc. | KOAI171017JW1 |
| HEK293T cells | Prof. Dr. Deyin Guo, Sun Yat-sen University, China | N/A |
| HEK293-TLR3 cells | Prof. Dr. Deyin Guo, Sun Yat-sen University, China | N/A |
| A549 cells | Prof. Dr. Deyin Guo, Sun Yat-sen University, China | N/A |
| HeLa cells | Prof. Dr. Deyin Guo, Sun Yat-sen University, China | N/A |
| THP-1 cells | Prof. Dr. Deyin Guo, Sun Yat-sen University, China | N/A |
| Mouse RAW264.7 cells | Prof. Dr. Deyin Guo, Sun Yat-sen University, China | N/A |
| DYRK4-KO HEK293T | This study | N/A |

| Reagent/resource | Reference or source | Identifier or catalog number |
|---|---|---|
| DYRK4-KD A549 cells | This study | N/A |
| TRIM71-KO HEK293T | This study | N/A |
| BMDCs | This study | N/A |
| BMDMs | This study | N/A |
| MLFs | This study | N/A |
| SeV | Prof. Dr. Hong-Bing Shu, Wuhan University, China | N/A |
| VSV | Prof. Dr. Bo Zhong, Wuhan University, China | N/A |
| VSV-GFP | Prof. Dr. Hong-Bing Shu, Wuhan University, China | N/A |
| HSV-1 | Prof. Dr. Hong-Bing Shu, Wuhan University, China | N/A |
| **Recombinant DNA** | | |
| pEF-Flag | This study | N/A |
| pEF-myc | This study | N/A |
| pEF-DYRK4-Flag | This study | N/A |
| pEF-myc-DYRK4 | This study | N/A |
| pEF-DYRK4-N-Flag | This study | N/A |
| pEF-DYRK4-NK-Flag | This study | N/A |
| pEF-DYRK4-K-Flag | This study | N/A |
| pEF-DYRK4-KC-Flag | This study | N/A |
| pEF-DYRK4-C-Flag | This study | N/A |
| pEF-DYRK4-NC-Flag | This study | N/A |
| pET30a-DYRK4-his | This study | N/A |
| pEF-DYRK4-K133A-Flag | This study | N/A |
| pEF-DYRK4-K133R-Flag | This study | N/A |
| pEF-DYRK4-Y264A-Flag | This study | N/A |
| pEF-DYRK4-Y264F-Flag | This study | N/A |
| pEF-DYRK4-K133A/Y264A-Flag | This study | N/A |
| pEF-DYRK4-K133R/Y264F-Flag | This study | N/A |
| pLKO.1-shDYRK4#5 | This study | N/A |
| pLKO.1-shDYRK4#6 | This study | N/A |
| PEF-DYRK4-GFP | This study | N/A |
| pRK-Flag | Prof. Dr. Hong-Bing Shu, Wuhan University, China | N/A |

| Reagent/resource | Reference or source | Identifier or catalog number |
| --- | --- | --- |
| pRK-Flag-RIG-I | Prof. Dr. Hong-Bing Shu, Wuhan University, China | N/A |
| pRK-Flag-MAVS | Prof. Dr. Hong-Bing Shu, Wuhan University, China | N/A |
| pRK-Flag-TBK1 | Prof. Dr. Hong-Bing Shu, Wuhan University, China | N/A |
| pRK-Flag-IRF3 | Prof. Dr. Hong-Bing Shu, Wuhan University, China | N/A |
| pRK-Flag-IRF3-5D | Prof. Dr. Hong-Bing Shu, Wuhan University, China | N/A |
| pCS2-Flag-IRF3-2A | This study | N/A |
| pCMV-Flag | This study | N/A |
| pCMV-Flag-cGAS | This study | N/A |
| pCMV-Flag-STING | This study | N/A |
| pRK-HA-Ub | Prof. Dr. Hong-Bing Shu, Wuhan University, China | N/A |
| pRK-HA-Ub-K0 | Prof. Dr. Hong-Bing Shu, Wuhan University, China | N/A |
| pRK-HA-Ub-K6 | Prof. Dr. Hong-Bing Shu, Wuhan University, China | N/A |
| pRK-HA-Ub-K11 | Prof. Dr. Hong-Bing Shu, Wuhan University, China | N/A |
| pRK-HA-Ub-K27 | Prof. Dr. Hong-Bing Shu, Wuhan University, China | N/A |
| pRK-HA-Ub-K29 | Prof. Dr. Hong-Bing Shu, Wuhan University, China | N/A |
| pRK-HA-Ub-K33 | Prof. Dr. Hong-Bing Shu, Wuhan University, China | N/A |
| pRK-HA-Ub-K48 | Prof. Dr. Hong-Bing Shu, Wuhan University, China | N/A |
| pRK-HA-Ub-K63 | Prof. Dr. Hong-Bing Shu, Wuhan University, China | N/A |
| pRK-HA-Ub-K6R | Prof. Dr. Hong-Bing Shu, Wuhan University, China | N/A |
| pRK-HA-Ub-K11R | Prof. Dr. Hong-Bing Shu, Wuhan University, China | N/A |
| pRK-HA-Ub-K27R | Prof. Dr. Hong-Bing Shu, Wuhan University, China | N/A |
| pRK-HA-Ub-K29R | Prof. Dr. Hong-Bing Shu, Wuhan University, China | N/A |

| Reagent/resource | Reference or source | Identifier or catalog number |
| --- | --- | --- |
| pRK-HA-Ub-K33R | Prof. Dr. Hong-Bing Shu, Wuhan University, China | N/A |
| pRK-HA-Ub-K48R | Prof. Dr. Hong-Bing Shu, Wuhan University, China | N/A |
| pRK-HA-Ub-K63R | Prof. Dr. Hong-Bing Shu, Wuhan University, China | N/A |
| PEF-IRF3-mcherry | This study | N/A |
| pGEX6P-1-GST | This study | N/A |
| pGEX6P-1-GST-IRF3 | This study | N/A |
| pRK-HA | This study | N/A |
| pRK-HA-IRF3 | This study | N/A |
| pCS2-Flag | This study | N/A |
| pCS2-Flag-IRF3 | This study | N/A |
| pCS2-Flag-IRF3-K5R | This study | N/A |
| pCS2-Flag-IRF3-K29R | This study | N/A |
| pCS2-Flag-IRF3-K39R | This study | N/A |
| pCS2-Flag-IRF3-K70R | This study | N/A |
| pCS2-Flag-IRF3-K77R | This study | N/A |
| pCS2-Flag-IRF3-K87R | This study | N/A |
| pCS2-Flag-IRF3-K98R | This study | N/A |
| pCS2-Flag-IRF3-K105R | This study | N/A |
| pCS2-Flag-IRF3-K193R | This study | N/A |
| pCS2-Flag-IRF3-K313R | This study | N/A |
| pCS2-Flag-IRF3-K315R | This study | N/A |
| pCS2-Flag-IRF3-K360R | This study | N/A |
| pCS2-Flag-IRF3-K366R | This study | N/A |
| pCS2-Flag-IRF3-K409R | This study | N/A |
| pCS2-Flag-IRF3-K87R/K105R | This study | N/A |
| pCS2-Flag-IRF3-DP | This study | N/A |
| pCS2-Flag-IRF3-DPI | This study | N/A |
| pCS2-Flag-IRF3-IR | This study | N/A |
| pCS2-Flag-IRF3-I | This study | N/A |
| pLKO.1-TCR-shNC | This study | N/A |
| pLKO.1-shTRIM71#1 | This study | N/A |

| Reagent/resource | Reference or source | Identifier or catalog number |
|---|---|---|
| pLKO.1-shTRIM71#2 | This study | N/A |
| pLKO.1-shTRIM71#3 | This study | N/A |
| pEF-TRIM71-Flag | This study | N/A |
| pEF-TRIM71-ΔRING-Flag | This study | N/A |
| pEF-TRIM71-ΔNHL-Flag | This study | N/A |
| pEF-TRIM71-FNHL-Flag | This study | N/A |
| pEF-TRIM71-NHL-Flag | This study | N/A |
| pEF-myc-TRIM71 | This study | N/A |
| pEF-myc-TRIM71-C12A/C15A | This study | N/A |
| pEF-myc-TRIM71-ΔRING | This study | N/A |
| pEF-TRIM4-Flag | This study | N/A |
| pEF-TRIM21-Flag | This study | N/A |
| pEF-TRIM23-Flag | This study | N/A |
| pEF-RNF138-Flag | This study | N/A |
| pEF-HOIP-Flag | This study | N/A |
| pEF-HOIL-1L-Flag | This study | N/A |
| pEF-Sharpin-Flag | This study | N/A |
| pGL3.0-IFN-β-lucifer | Prof. Dr. Hong-Bing Shu, Wuhan University, China | N/A |
| pGL3.0-ISRE-lucifer | Prof. Dr. Hong-Bing Shu, Wuhan University, China | N/A |
| pGL3.0-NF-κB-lucifer | Prof. Dr. Hong-Bing Shu, Wuhan University, China | N/A |
| pRL-TK | Prof. Dr. Hong-Bing Shu, Wuhan University, China | N/A |
| **Antibodies** | | |
| Mouse anti-Dyrk4 | Santa Cruz | #sc-393479 |
| Mouse anti-TRIM71 | Santa Cruz | #sc-393352 |
| Rabbit anti-IRF3 | Proteintch | #11312-1-AP |
| Rabbit anti-Lamin B | Proteintch | #12987-1-AP |
| Rabbit anti-GAPDH | Proteintch | #10494-1-AP |
| Rabbit anti-GFP | Proteintch | #50430-2-AP |
| Rabbit anti-Flag | Proteintch | #20543-1-AP |
| Rabbit anti-myc | Proteintch | #16286-1-AP |
| Rabbit anti-HA | Proteintch | #51064-2-AP |
| Rabbit anti-GST | Proteintch | #10000-0-AP |
| Mouse anti-His | Proteintch | #66005-1-Ig |
| Mouse anti-CD11b | Proteintch | #FITC-65055 |
| Mouse anti-CD11c | Proteintch | #APC-65130 |

| Reagent/resource | Reference or source | Identifier or catalog number |
|---|---|---|
| Mouse anti-IgG | Proteintch | #B900620 |
| Mouse anti-β-actin | Proteintch | #66009-1-Ig |
| Rabbit anti-p-p65 | CST | #3033 |
| Rabbit anti-p65 | CST | #8242 |
| Rabbit anti-p-IRF3(S396) | CST | #29047 |
| Rabbit anti-p-IRF3(S386) | Huabio | #ET1608-22 |
| Rabbit anti-p-IκBα(S32) | abcam | #ab92700 |
| Rabbit anti-IκBα | abcam | #ab32518 |
| Rabbit anti-Ubiquitin | ABclonal | #A0162 |
| Rabbit anti-DYRK4 | This study | Polyclonal antibody to human DYRK4 was elicited by immunization of rabbit with the 401-520aa peptide of DYRK4 and was purified by ion-exchange chromatography |
| **Oligonucleotides and other sequence-based reagents** | | |
| Primers for sgRNA | This study | Appendix Table S1 |
| Primers for qRT-PCR | This study | Appendix Table S2 |
| Primers for shRNA | This study | Appendix Table S3 |
| **Chemicals, enzymes, and other reagents** | | |
| poly(I:C) HMW | Invivogen | tlrl-pic |
| LPS | ThermoFisher | 00-4976-93 |
| IFN-γ | Peprotech | #300-02-1MG |
| Cycloheximide (CHX) | Aladdin | 66-81-9 |
| MG132 | Sigma | 1211877-36-9 |
| PrimeScript™ RT reagent Kit with gDNA Eraser | Takara | RR047A |
| Hieff qRT-PCR SYBR Green Master Mix | Yeasen | CAT#11203ES08 |
| 100× PMSF | TargetMol | 329-98-6 |
| Protease Inhibitor Cocktail (100×) | TargetMol | C0001 |
| Phosphatase Inhibitor Cocktail (100×) | TargetMol | C0003 |
| Fetal bovine serum (FBS) | BI | 04-001-1ACS |
| T4 DNA Ligase | Vazyme | C301-01 |
| Phanta Max Super-Fidelity DNA Polymerase | Vazyme | P505-d1 |
| Anti-Flag beads | Yeasen | 20584ES25 |
| protein A/G beads | Biolinkedin | L-2104 |
| NC membranes | Merck Millipore | N/A |
| ECL | Yeasen | 36208ES76 |

| Reagent/resource | Reference or source | Identifier or catalog number |
|---|---|---|
| Dual-specific luciferase reporter assay system | Promega | E1980 |
| DAPI | Wuhan Promoter Biology Co., Ltd. | N/A |
| TRIzol | Invitrogen | N/A |
| GM-CSF | Peprotech | N/A |
| M-CSF | Peprotech | N/A |
| type II collagenase | BioFroxx | N/A |
| DNase I | Sigma-Aldrich | N/A |
| **Software** | | |
| GraphPad Prism | https://www.graphpad.com | N/A |
| Primer Premier 5 | N/A | N/A |
| Photoshop | N/A | N/A |
| Adobe Illustrator 2024 | N/A | N/A |
| ImageJ | https://imagej.nih.gov/ij/index.html | N/A |
| **Other** | | |
| Endo-free Plasmid Mini Kit | Omega | D6950-02 |
| Gel Extraction Kit | Omega | D2500-02 |
| Nuclear-cytoplasmic extraction kit | Wuhan KeRui Biotechnology Co., Ltd. | N/A |
| Histological analysis | Baiqiandu Biology Co., Ltd. | N/A |

## Mice

$Dyrk4^{-/-}$ mice were generated via CRISPR/Cas9-mediated genome editing (Cyagen Biosciences, Inc.). In brief, the vectors encoding Cas9 (44758, Addgene) and guide RNAs (5'-CAGAT-CATGGGCATTAAGGCTGG-3'; 5'-GAATGAAGGTAATAATGC CGTGG-3') were transcribed into mRNAs in vitro, and then, the gRNAs were injected into fertilized eggs, which were subsequently transplanted into pseudopregnant mice. The target genome of F0 mice was amplified and sequenced via PCR, and the chimeras were crossed with wild-type C57BL/6 mice to obtain $Dyrk4^{+/-}$ mice. F1 $Dyrk4^{+/-}$ mice (including one male and two females) were crossed to obtain $Dyrk4^{+/+}$ and $Dyrk4^{-/-}$ mice via PCR analysis. We used 8-week-old and sex-matched $Dyrk4^{+/+}$ and $Dyrk4^{-/-}$ littermate mice for the experiments. All the mice were housed in specific pathogen-free (SPF) animal facilities at Wuhan University, and the animal experiments were conducted under the supervision of the Institutional Animal Care and Use Committee of Wuhan University with the approval number SKLV-AE2021 002.

## Cell culture

HEK293T, HEK293-TLR3, A549, HeLa, and RAW264.7 cells were cultured in DMEM supplemented with 10% FBS and 1% penicillin/streptomycin. THP-1 cells were cultured in RPMI-1640 supplemented with 10% FBS and 1% penicillin/streptomycin and treated with PMA (100 ng/ml) for 3 days before viral infection. Bone marrow-derived dendritic cells (BMDCs) and bone marrow-derived macrophages (BMDMs) were isolated from the femurs and tibias of 8-week-old $Dyrk4^{+/+}$ and $Dyrk4^{-/-}$ mice and cultured for 7 days in complete DMEM containing GM-CSF (50 ng/ml) and M-CSF (50 ng/ml) cytokines, respectively. Mouse lung fibroblasts (MLFs) were isolated from the lungs of 8-week-old $Dyrk4^{+/+}$ and $Dyrk4^{-/-}$ mice, minced and digested in 1×HBSS buffer supplemented with 10 mg/ml type II collagenase and 0.1% DNase I for 4 h at 37 °C with shaking. The cell suspensions were filtered through the sterile mesh, and the filtered cells were cultured in DMDM supplemented with 10% FBS and 1% penicillin/streptomycin. Two days later, adherent MLFs were rinsed with HBSS and cultured for experiments.

## Viral infection

For the qPCR analysis, the cells were seeded into 12-well plates ($3-5 \times 10^5$ cells per well) and infected with SeV or HSV-1 (MOI of 10) for the indicated durations. For viral replication by fluorescence microscopy analysis, cells were seeded into 6-well plates ($1 \times 10^6$ cells per well) and infected with VSV-GFP (0.1 MOI) for the indicated durations. For infection, age- and sex-matched $Dyrk4^{+/+}$ and $Dyrk4^{-/-}$ mice were injected with VSV ($1 \times 10^8$ PFU/per mouse), respectively, and the survival of the animals was monitored every day. The lungs, spleens or livers were collected for qPCR analysis at 24 h after infection.

## RNA purification and qRT-PCR

Total RNA was extracted from cells via TRIzol. A reverse transcription system was used to synthesize cDNA. Hieff qRT-PCR SYBR Green Master Mix and a Bio-Rad CFX Connect system were used for qPCR. The mRNA results were normalized to GAPDH expression. The qPCR sequences of primers used in this study are listed in Appendix Table S2 (see Reagents and Tools Table).

## shRNA

The shRNAs targeting DYRK4 or TRIM71 genes were constructed via the pLKO.1-TCR cloning vector and transfected with PEI into cells, followed by immunoblotting or qPCR analysis. The shRNA sequences of primers used in this study are listed in Appendix Table S3 (see Reagents and Tools Table).

## Generation of knockout cells via CRISPR/Cas9 technology

HEK293T cells with knockout of DYRK4 or TRIM71 and A549 cells with knockdown of DYRK4 were generated via a lenti-CRISPR/Cas9-v2 system. HEK293T cells were seeded in 100 mm

dishes and transfected with lenti-CRISPR-DYRK4-sgRNA#1, sgRNA#2 or lenti-CRISPR-TRIM71-sgRNA along with the packaging plasmids pMD2. G and psPAX2 using PEI transfection reagent. The medium was changed 8 h after transfection. The supernatants containing lentivirus were harvested 72 h after infection and were filtered through a 0.45-μm filter. HEK293T and A549 cells were incubated with lentivirus for 48 h and then selected with puromycin for 5–7 days. The isolation of single clonal knockout or knockdown cells was confirmed via western blotting. The sgRNA sequences of primers used in this study are listed in Appendix Table S1 (see Reagents and Tools Table).

## MS

HEK293T cells were transfected with control and DYRK4-Flag for 24 h, respectively, then, the cells were lysed, subsequently purified with Flag beads, separated by SDS-PAGE, and stained with Coomassie blue. The deeper color protein band was cut from the gel compared with the control bands, followed by mass spectrometry (MS) analysis.

## Coimmunoprecipitation and immunoblot analysis

For coimmunoprecipitation (co-IP) assays, the cells were collected and lysed on ice for 15 min with 800 μl of NP-40 lysis buffer (50 mM Tris-HCl, pH 7.4; 150 mM NaCl; 2 mM EDTA; 1% NP-40; 10 mM NaF; 1 mM Na3VO4; and 2 mM DTT) containing protease inhibitors (Sigma). After centrifugation at 12,000 rpm for 10 min at 4 °C, the supernatants were collected and incubated with anti-Flag beads or the indicated antibodies plus protein A/G beads. After incubation overnight at 4 °C, the beads were washed three times with wash buffer. The immunoprecipitates were eluted by boiling with a 2× SDS loading buffer and subjected to immunoblot analysis. For immunoblot analysis, whole-cell extracts were collected and lysed on ice for 15 min with NP-40 lysis buffer containing protease inhibitors. The lysates were subjected to SDS-PAGE, transferred onto NC membranes, incubated with the indicated antibodies, and then visualized by enhanced chemiluminescence (ECL) western blotting detection reagent. Images were taken with SYNGENE and processed with Adobe Photoshop.

## Protein purification and GST pull-down assay

The plasmids encoding pGEX6P-1-GST and pGEX6P-1-GST-IRF3 (full length) were transformed into BL21 competent cells, respectively, which were induced with IPTG (0.2 mM) at 16 °C for 20 h. The plasmids encoding pET30a-DYRK4-his (full length) were transformed into Rosetta-competent cells, which were induced with IPTG (0.2 mM) at 16 °C for 20 h. The purified GST or GST-IRF3 was incubated with DYRK4-his at 4 °C overnight, followed by glutathione agarose pull-down for 2 h in PBS containing protease inhibitors. The glutathione agarose beads were collected and washed three times with PBS and mixed with 2× SDS loading buffer, followed by immunoblot analysis.

## Ubiquitination assays

The cells were lysed on ice with NP-40 lysis buffer (100 μl, in the presence of 1% SDS), and the cell lysates were denatured at 100 °C for 5 min and then diluted to 10 volumes of lysis buffer before

incubation with either anti-Flag beads or protein A/G plus anti-IRF3 at 4 °C overnight. The immunoprecipitates were washed three times and analyzed by immunoblotting.

## Transfection and reporter gene assay

HEK293T-WT or KO cells were seeded in 24-well plates and then transfected with IFN-β-luc, ISRE-luc, NF-κB-luc, DYRK4 or TRIM71 or control vector, together with the pRL-TK Renilla luciferase reporter plasmid as a control. Twenty-four hours after transfection, the cells were infected with SeV for another 12 h. Luciferase activity in the cell lysates was measured with a dual-specific luciferase reporter assay system.

## ELISA

The mouse IFN-β and CXCL10 ELISA kits were obtained from Jiangsu Jingmei Biotechnology Co., Ltd. and were used for the detection of proteins in mouse serum.

## Native PAGE

Native polyacrylamide gels were prepared from 8% acrylamide gels without SDS. The gel was prerun for 30 min at 40 mA on ice with running buffer (25 mM Tris-HCl, pH 8.4, and 192 mM glycine with or without 0.5% deoxycholate in the cathode and anode buffers, respectively). The lysate (without SDS) was centrifuged at 12,000 rpm at 4 °C for 10 min to remove the insoluble fraction. The sample was mixed with 2× loading buffer (50 mM Tris-HCl, pH 6.8, 30% glycerol and bromophenol blue), applied to the gel and subjected to electrophoresis for 60 min at 35 mA on ice, followed by immunoblot analysis.

## Lung histology

Lungs from PBS-treated or virus-infected $Dyrk4^{+/+}$ and $Dyrk4^{-/-}$ mice were collected and fixed with 4% paraformaldehyde. The tissue samples were paraffin-embedded, stained with hematoxylin and eosin solution, and then subjected to light microscopy for histological analysis.

## Plaque assay

HEK293T cells were transfected with the control or DYRK4 plasmid for 24 h, mock-transfected or transfected with poly(I:C) (1 μg) for 16 h, and then infected with VSV-GFP (MOI of 0.1). The supernatants containing the virus were harvested 24 h after infection and diluted to infect Vero cells plated on 12-well plates at 90% confluence. After 2 h of infection at 37 °C, the supernatants were removed, the cells were washed three times with PBS, 3% methylcellulose was added, and the plates were incubated for 48 h. The overlay was removed, and the cells were fixed with 4% paraformaldehyde for 30 min and stained with crystal violet (0.2%) for 30 min. The plaques were counted, averaged and multiplied by the dilution factor to determine the virus titer as PFU/ml.

## Confocal microscopy

HeLa cells were seeded in confocal dishes and transfected with plasmids encoding DYRK4-GFP and IRF3-mCherry for 24 h, then

the cells were fixed in 4% paraformaldehyde for 30 min. The nuclei were stained with DAPI. Imaging of the cells was carried out via a Leica laser-scanning confocal microscope.

## Subcellular fractionation

Cytoplasmic and nuclear extracts were prepared with a nuclear-cytoplasmic extraction kit according to the manufacturer's instructions.

## Statistical analysis

All the results were analyzed using Microsoft Excel and GraphPad Prism 8.0. The results are representative of three biological replicate experiments and are shown as the means with SEMs, as indicated in the figure legends. We used the two-tailed Student's $t$ test to calculate the statistical significance. $P$ values < 0.05 were considered statistically significant. For mouse survival analysis, Kaplan–Meier survival curves were generated and analyzed for statistical significance via GraphPad Prism 8.0. The researchers were not blinded during data collection and analysis.

# Data availability

This study includes no data deposited in external repositories.

The source data of this paper are collected in the following database record: biostudies:S-SCDT-10_1038-S44319-024-00352-x.

# Peer review information

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

## Acknowledgements

This work was supported by the National Natural Science Foundation of China (Grant Nos. 31370187, 30870113, 81572447, 31871427, and 32260173), Zhongnan Hospital—Taikang Medical School (School of Basic Medical Sciences) of Wuhan University Joint Foundation (Grant No. JCZN2022010), and the Central Guidance on Local Science and Technology Development Fund of Tibet (Grant No. XZ202301YD0040C). The authors thank Dr. Deyin Guo, Dr. Chao Wu (Sun Yat-sen University), Dr. Hong-Bing Shu, and Dr. Bo Zhong (Wuhan University) for their help. The authors thank the Core Facilities for Wuhan University Medical Structural Biology Research Center for helping with the confocal microscopy analysis.

## Author contributions

**Xianhuang Zeng**: Conceptualization; Data curation; Software; Validation; Investigation; Methodology; Writing—original draft. **Jiaqi Xu**: Software; Formal analysis; Methodology. **Jiaqi Liu**: Software; Formal analysis; Methodology. **Yang Liu**: Formal analysis; Validation; Investigation. **Siqi Yang**: Software; Validation; Investigation. **Junsong Huang**: Resources; Software; Validation; Investigation. **Chengpeng Fan**: Resources; Software; Formal analysis; Investigation. **Mingxiong Guo**: Conceptualization; Supervision; Funding acquisition; Writing—review and editing. **Guihong Sun**: Conceptualization; Supervision; Funding acquisition; Writing—review and editing.

Source data underlying figure panels in this paper may have individual authorship assigned. Where available, figure panel/source data authorship is listed in the following database record: biostudies:S-SCDT-10_1038-S44319-024-00352-x.

## Disclosure and competing interests statement

The authors declare no competing interests.

# Expanded View Figures

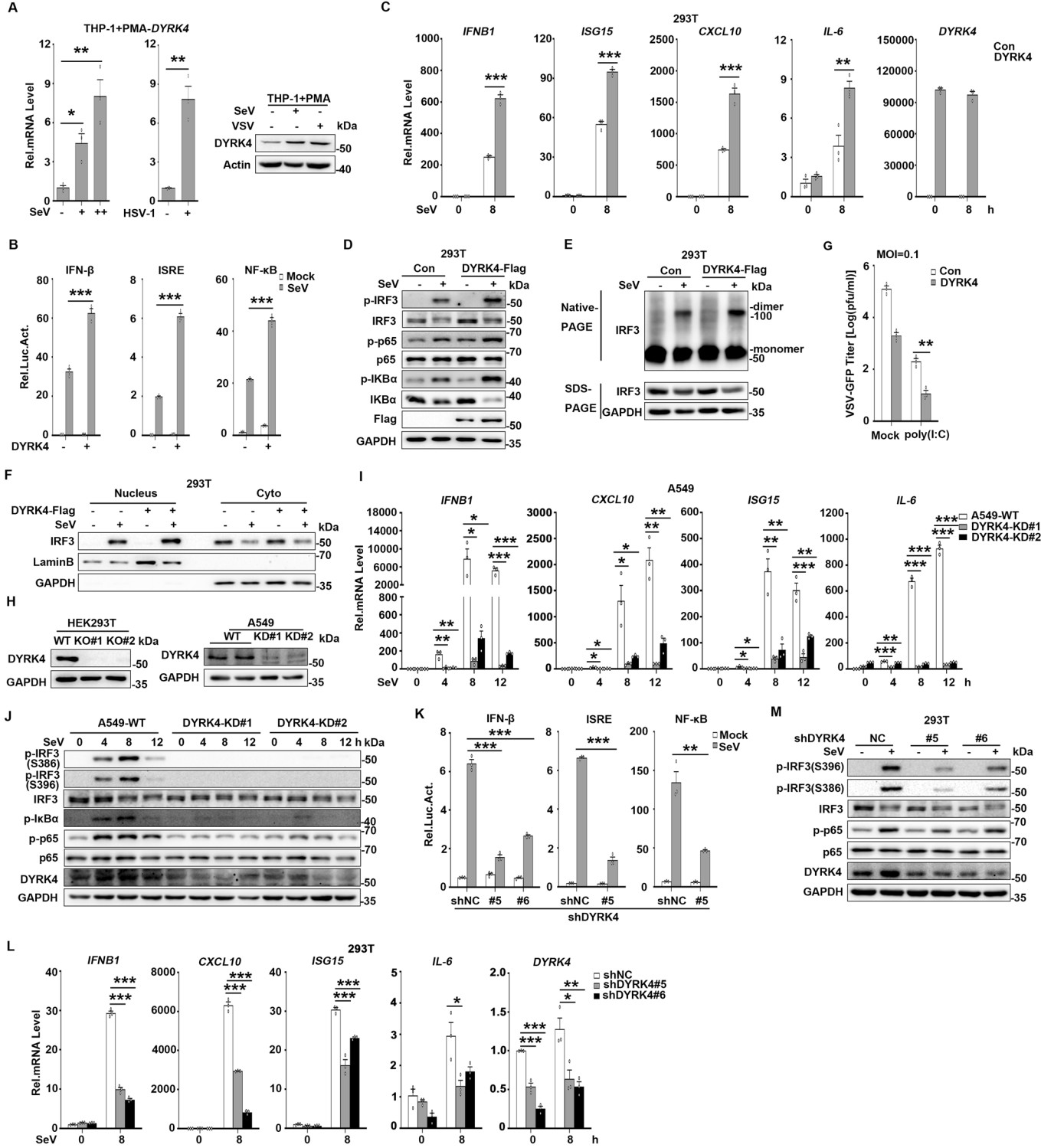

◀ **Figure EV1. DYRK4 overexpression or knockdown promotes or inhibits virus-triggered signaling.**

Effects of viral infection on DYRK4 mRNA and protein levels. THP-1-derived macrophages were infected with SeV or HSV-1 for 8 h before qPCR analysis and were infected with SeV or VSV for 8 h before immunoblotting analysis with anti-DYRK4 (right). (P value; SeV +: 0.01049, SeV ++: 0.00497, HSV-1: 0.00216) (n = 3 biological replicates). (**A**) DYRK4 activated the IFN-β promoter, ISRE, and NF-κB. HEK293T cells were transfected with the IFN-β, ISRE, and NF-κB reporters and the control or DYRK4 plasmid for 24 h and then infected with SeV for 12 h before luciferase assays. (P value; IFN-β: 0.00036, ISRE: $1.1 \times 10^{-5}$, NF-κB: $4.0 \times 10^{-5}$) (n = 3 biological replicates). (**B**) Effects of DYRK4 on the SeV-induced transcription of downstream genes. HEK293T cells were transfected with the control or DYRK4 plasmid for 24 h and then infected with SeV for 8 h before qPCR analysis. (P value; IFNB1: $7.6 \times 10^{-5}$, CXCL10: 0.00058, ISG15: 0.000143, IL-6: 0.0097) (n = 3 biological replicates). (**C**) Effects of DYRK4 on SeV-induced phosphorylation of IRF3 (Ser396), p65 and IκBα. HEK293T cells were transfected with the control or DYRK4 plasmid for 24 h and then infected with SeV for 8 h before immunoblotting analysis with the indicated antibodies. (**D**) DYRK4 enhanced the SeV-induced dimerization of IRF3. HEK293T cells were transfected with the control or DYRK4 plasmid for 24 h and then infected with SeV for 8 h. Cell lysates were separated by native (upper panel) or SDS (bottom panel) PAGE and analyzed by immunoblotting with the indicated antibodies. (**E**) Effects of DYRK4 on the SeV-induced nuclear translocation of IRF3. HEK293T cells were transfected with the control or DYRK4 plasmid for 24 h and then infected with SeV for 8 h. Immunoblot analysis of IRF3 in the cytoplasmic (Cyto) and nuclear fractions was performed with the indicated antibodies. (**F**) Effects of DYRK4 on VSV-GFP replication. HEK293T cells were transfected with the control or DYRK4 plasmid for 24 h, mock-transfected or transfected with poly(I:C) (1 μg) for 16 h, and then infected with VSV-GFP (MOI of 0.1). The supernatants were harvested 24 h after infection for standard plaque assays. (P value = 0.00177) (n = 3 biological replicates). (**G**) DYRK4-deficient (KO) HEK293T clones and DYRK4-knockdown (KD) A549 cells were generated via the CRISPR-Cas9 method. Deficiencies of DYRK4 in the KO clones and DYRK4-KD A549 cells were confirmed by immunoblotting analysis with anti-DYRK4. (**H**) Effects of DYRK4 knockdown on the SeV-induced transcription of downstream genes. DYRK4-KD and control A549 cells were infected with SeV for the indicated times before qPCR analysis. (P value; IFNB1; 4 h; KD#1: 0.0053, KD#2: 0.0048, 8 h; KD#1: 0.0228, KD#2: 0.0254, 12 h; KD#1: 0.00065, KD#2: 0.00072, CXCL10; 4 h; KD#1: 0.0259, KD#2: 0.0307, 8 h; KD#1: 0.014, KD#2: 0.0213, 12 h; KD#1: 0.00106, KD#2: 0.0033, ISG15; 4 h; KD#1: 0.048, KD#2: 0.044, 8 h; KD#1: 0.0023, KD#2: 0.0048, 12 h; KD#1: 0.00096, KD#2: 0.0029, IL-6; 4 h; KD#1: $2.1 \times 10^{-5}$, KD#2: 0.0079, 8 h; KD#1: $9.3 \times 10^{-6}$, KD#2: $1.1 \times 10^{-5}$, 12 h; KD#1: $4.2 \times 10^{-6}$, KD#2: $4.6 \times 10^{-6}$) (n = 3 biological replicates). (**I**) Effects of DYRK4 knockdown on SeV-induced phosphorylation of IRF3 (Ser396, Ser386), p65 and IκBα. DYRK4-KD and control A549 cells were infected with SeV for the indicated times before immunoblotting analysis with the indicated antibodies. (**J**) Effects of DYRK4 knockdown on Sev-induced activation of the IFN-β promoter, ISRE, and NF-κB. HEK293T cells were transfected with the IFN-β, ISRE, NF-κB reporter, and control or DYRK4 RNAi plasmids for 36 h and then infected with SeV for 12 h before luciferase assays. (P value; IFN-β; #5: $3.6 \times 10^{-5}$, #6: $7.0 \times 10^{-5}$, ISRE; #5: $5.1 \times 10^{-6}$, NF-κB; #5: 0.00305) (n = 3 biological replicates). (**K**) Effects of DYRK4 knockdown on the SeV-induced transcription of downstream genes. HEK293T cells were transfected with control or DYRK4 RNAi plasmids for 36 h, after which the cells were infected with SeV for 8 h before qPCR analysis. (P value; IFNB1; #5: $1.2 \times 10^{-5}$, #6: $4.0 \times 10^{-6}$, CXCL10; #5: $4.5 \times 10^{-5}$, #6: $7.9 \times 10^{-6}$, ISG15; #5: 0.00065, #6: 0.00023, IL-6; #5: 0.0272, DYRK4; 0 h; #5: 0.00057, #6: $1.9 \times 10^{-5}$, 8 h; #5: 0.0245, #6: 0.0089) (n = 3 biological replicates). (**L**) Effects of DYRK4 knockdown on SeV-induced phosphorylation of IRF3 (Ser396, Ser386) and p65. HEK293T cells were transfected with control or DYRK4 RNAi plasmids for 36 h and then infected with SeV for 8 h before immunoblotting analysis with the indicated antibodies. Data information: Data are representative of three biological replicates and are shown as the means with SEMs (**A–C, G, I, K, L**); data in (**D–F, H, J, M**) are representative of two replicates. *P < 0.05, **P < 0.01, ***P < 0.001; two-tailed unpaired Student's t test.

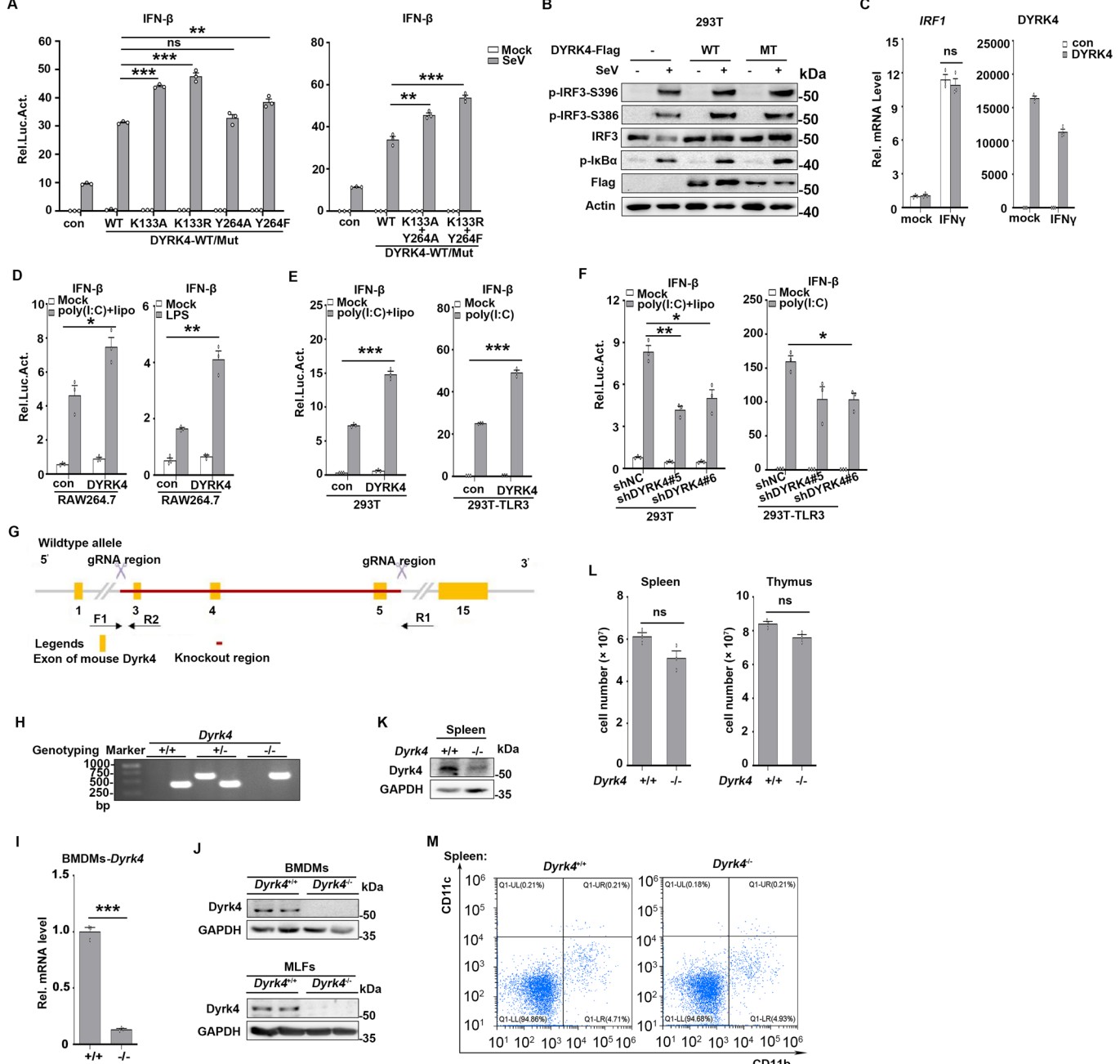

◀ **Figure EV2. DYRK4 promotes virus-triggered signaling independent of its kinase activity and the generation and analysis of Dyrk4-knockout mice.**

DYRK4 and its mutant (left) or double mutants (right) potentiate SeV-induced activation of the IFN-β promoter. The experiments were performed as described in Fig. S1B. (P value; K133A: $8.8 \times 10^{-6}$, K133R: 0.00015, Y264A: ns, Y264F: 0.00302, K133A + Y264A: $4.4 \times 10^{-6}$, K133R + Y264F: $3.3 \times 10^{-6}$) ($n = 3$ biological replicates). (**A**) Effects of DYRK4-MT on SeV-induced phosphorylation of IRF3 (Ser396, Ser386) and IκBα. HEK293T cells were transfected with control or DYRK4-WT or DYRK4-MT (K133R/ Y264F) plasmids for 24 h and then infected with SeV for 8 h before immunoblotting analysis with the indicated antibodies. (**B**) DYRK4 does not potentiate the IFN-γ-induced transcription of *IRF1*. HEK293T cells were transfected with the control or DYRK4 plasmid for 24 h and then treated with IFN-γ (100 μg/ml) for 8 h before qPCR analysis. (ns: not significant) ($n = 3$ biological replicates). (**C**) Effects of DYRK4 on cytoplasmic poly(I:C)-induced or LPS-induced activation of the IFN-β promoter. RAW264.7 cells were transfected with the IFN-β reporter and control or DYRK4 plasmid for 24 h, mock-transfected or transfected with poly(I:C) (5 μg) with Lipofectamine 2000, and untreated or treated with LPS (10 μg/ml) for 12 h before luciferase assays. (P value; poly(I:C): 0.0229, LPS: 0.0012) ($n = 3$ biological replicates). (**D**) Effects of DYRK4 on cytoplasmic poly(I:C)-induced or TLR3-mediated activation of the IFN-β promoter. HEK293T or HEK293T-TLR3 cells were transfected with the IFN-β reporter and control or DYRK4 plasmid for 24 h, mock-transfected or transfected with poly(I:C) (5 μg) with Lipofectamine 2000, and untreated or treated with poly(I:C) (25 μg/ml) for 12 h before luciferase assays. (P value; 293 T: $7.3 \times 10^{-5}$, 293T-TLR3: $2.4 \times 10^{-5}$) ($n = 3$ biological replicates). (**E**) Effects of DYRK4 knockdown on cytoplasmic poly(I:C)-induced or TLR3-mediated activation of the IFN-β promoter. HEK293T or HEK293T-TLR3 cells were transfected with the IFN-β reporter and control or DYRK4 RNAi plasmids for 36 h, mock-transfected or transfected with poly(I:C) (5 μg) with Lipofectamine 2000, and untreated or treated with poly(I:C) (25 μg/ml) for 12 h before luciferase assays. (P value; 293 T; #5: 0.00128, #6: 0.011, 293T-TLR3; #6: 0.0103) ($n = 3$ biological replicates). (**G**) A scheme for CRISPR/Cas9-mediated genome editing of the Dyrk4 gene locus. (**H**) Genotyping analysis of *Dyrk4*$^{+/+}$, *Dyrk4*$^{+/-}$ and *Dyrk4*$^{-/-}$ mice. (**I–K**) qPCR and immunoblot analysis of Dyrk4 mRNA or protein levels in *Dyrk4*$^{+/+}$ and *Dyrk4*$^{-/-}$ BMDMs, MLFs and spleens. (P value; I: $2.2 \times 10^{-5}$, L: $2.4 \times 10^{-5}$) ($n = 3$ biological replicates). (**L**) Cell counts in the spleen and thymus of *Dyrk4*$^{+/+}$ and *Dyrk4*$^{-/-}$ mice. (ns: not significant) ($n = 3$ biological replicates). (**M**) Flow cytometry analysis of the percentages of myeloid cells isolated from the spleens of *Dyrk4*$^{+/+}$ and *Dyrk4*$^{-/-}$ mice. Data information: Data are representative of three biological replicates and are shown as the means with SEMs (**A, C–F, I, L**); data in (**B, H, J, K, M**) are representative of two replicates. *$P < 0.05$, **$P < 0.01$, ***$P < 0.001$; two-tailed unpaired Student's $t$ test.

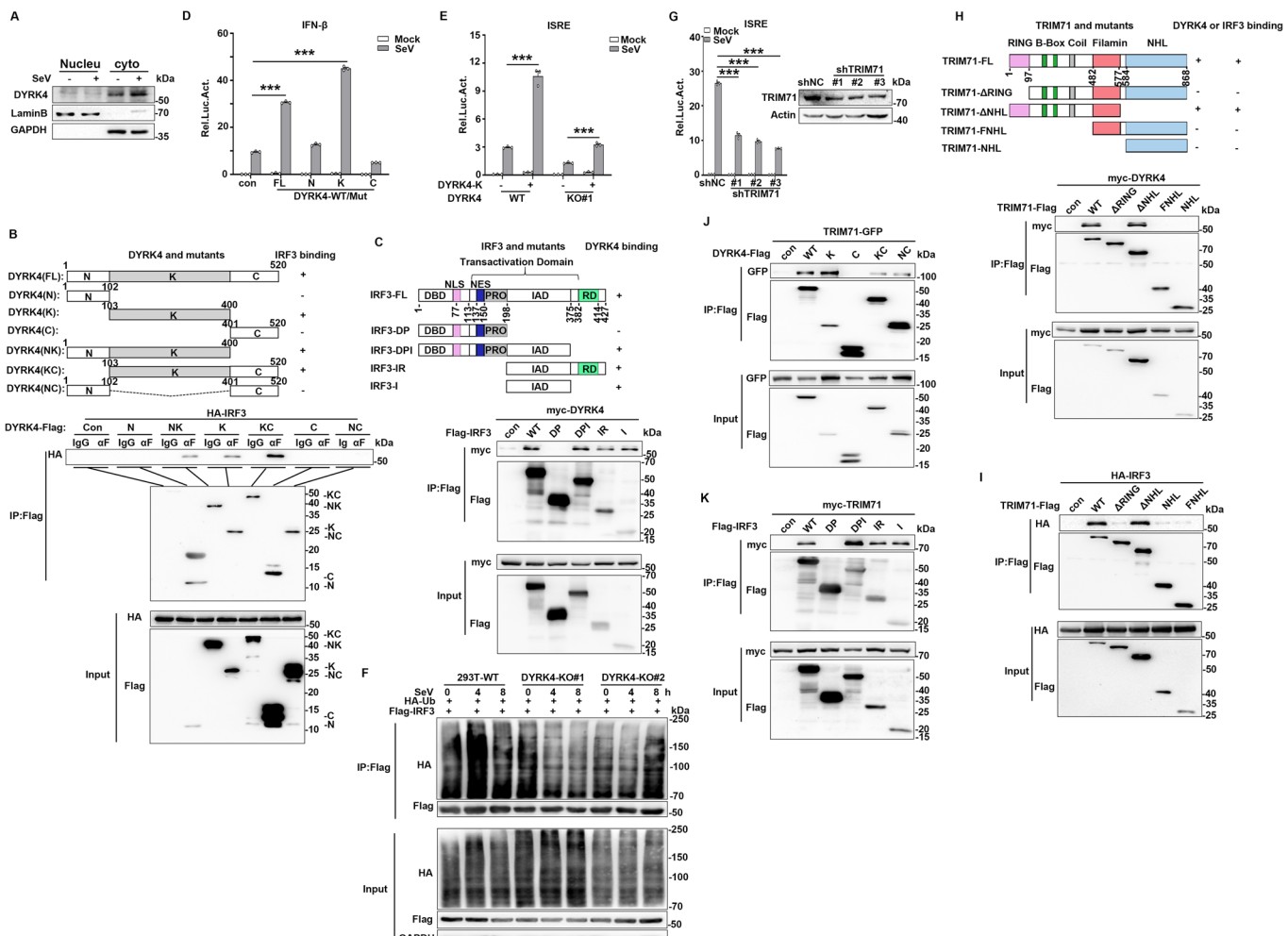

**Figure EV3. Domain mapping of the interaction of DYRK4 with TRIM71 or IRF3.**

Immunoblot analysis of DYRK4 in the cytoplasmic (Cyto) and nuclear (Nuc) fractions of HEK293T cells infected with SeV for 8 h. (B, C) Domain mapping of the interaction of DYRK4 with IRF3. HEK293T cells were transfected with the indicated truncations before coimmunoprecipitation and immunoblotting analysis with the indicated antibodies. Schematic representations of DYRK4 and IRF3 truncations are shown at the top. (DBD: DNA-binding structural domain, NLS: nuclear localization sequence, NES: nuclear export sequence, PRO: proline-rich region, IAD: IRF-related structural domain). (D) Effects of DYRK4 and its truncations on SeV-induced activation of the IFN-β promoter. HEK293T cells were transfected with the IFN-β reporter and the indicated plasmids for 24 h and then infected with SeV for 12 h before luciferase assays. ($P$ value; FL: $3.7 \times 10^{-7}$, K: $2.5 \times 10^{-7}$) ($n = 3$ biological replicates). (E) Effects of DYRK4-KO on SeV-induced activation of the IFN-β promoter. DYRK4-KO and control HEK293T cells were transfected with DYRK4-K truncation plasmids for 24 h and then infected with SeV for 12 h before luciferase assays. ($P$ value; WT: 0.00013, KO#1: 5.9 $\times 10^{-5}$) ($n = 3$ biological replicates). (F) Effects of DYRK4 deficiency on the SeV-induced polyubiquitination of IRF3. DYRK4-KO and control HEK293T cells were transfected with Flag-IRF3 and HA-Ub for 24 h and then infected with SeV for the indicated times before immunoblotting and coimmunoprecipitation analysis with the indicated antibodies. (G) Effects of TRIM71 knockdown on SeV-induced activation of ISRE. HEK293T cells were transfected with the ISRE reporter and TRIM71-RNAi plasmids for 36 h and then infected with SeV for 12 h before luciferase assays were performed, and the knockdown efficiency of TRIM71 was determined via immunoblot analysis. ($P$ value; #1: $9.6 \times 10^{-6}$, #2: $1.4 \times 10^{-6}$, #3: $2.9 \times 10^{-7}$) ($n = 3$ biological replicates). (H–K) Domain mapping of the interaction of TRIM71 with DYRK4 or IRF3. HEK293T cells were transfected with the indicated truncations for 24 h before immunoblotting and coimmunoprecipitation analysis with the indicated antibodies. Schematic representations of TRIM71 truncations are shown at the top. Data information: Data are representative of three biological replicates and are shown as the means with SEMs (D, E, G); data in (A–C, F, H–K) are representative of two replicates. ***$P < 0.001$, two-tailed unpaired Student's t test.

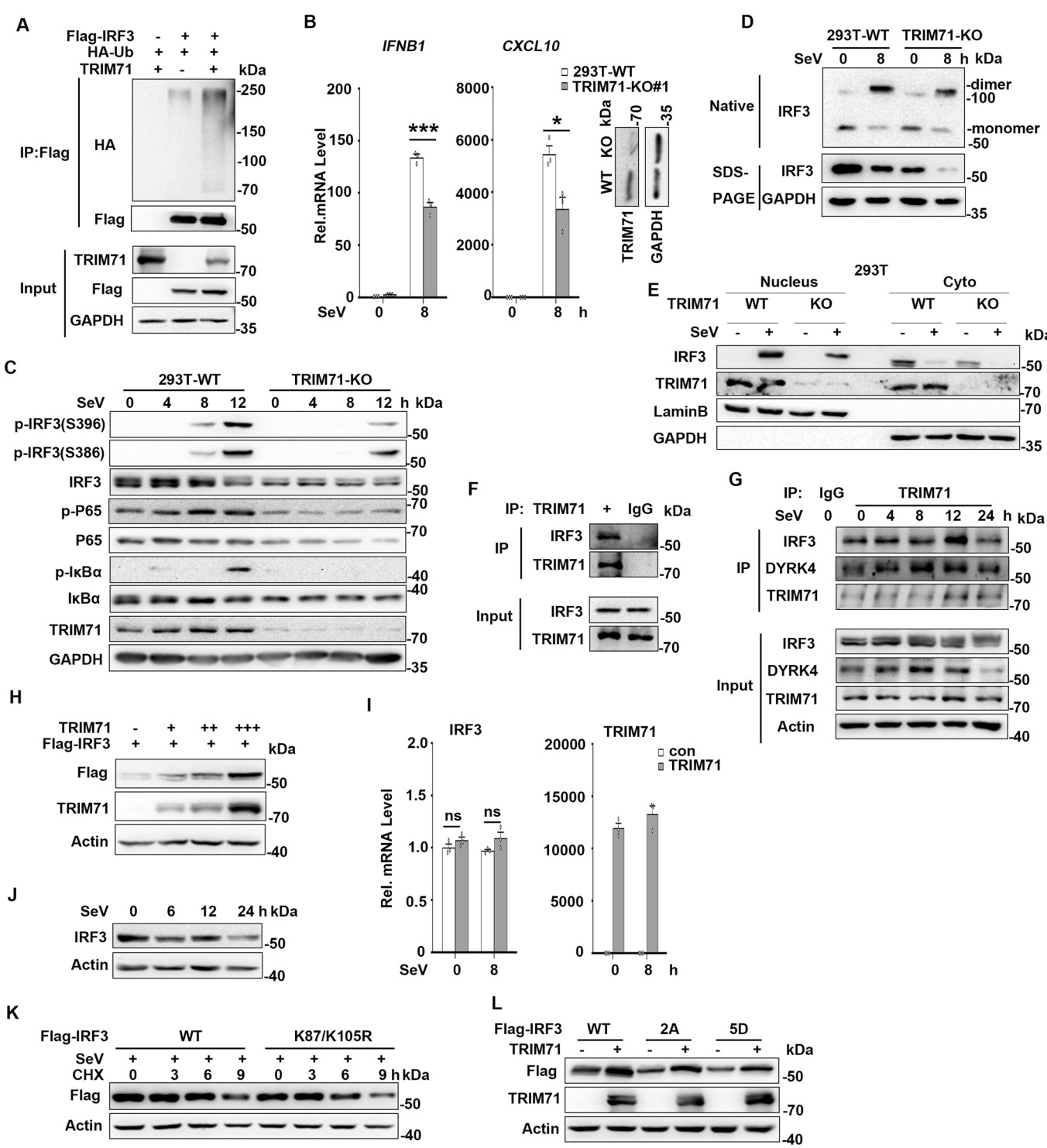

◀ **Figure EV4.  TRIM71 positively regulates RNA virus-triggered signaling by promoting IRF3 stability and activation.**

(A) Effects of TRIM71 on the polyubiquitination of IRF3. HEK293T cells were transfected with Flag-IRF3 or HA-Ub and control or TRIM71 plasmids for 24 h, followed by immunoblotting and coimmunoprecipitation analysis with the indicated antibodies. (B) Effects of TRIM71 deficiency on the SeV-induced transcription of downstream genes. TRIM71-KO HEK293T clones were generated via the CRISPR-Cas9 method. TRIM71-KO and control HEK293T cells were infected with SeV for 8 h before qPCR analysis, and the TRIM71 knockout efficiency was determined via immunoblot analysis. (*P* value; *IFNB1*: 0.00078, *CXCL10*: 0.01818) (*n* = 3 biological replicates). (C) Effects of TRIM71 deficiency on SeV-induced phosphorylation of IRF3 (Ser386, Ser396), p65 and IκBα. TRIM71-KO and control HEK293T cells were infected with SeV for the indicated times before immunoblotting analysis with the indicated antibodies. (D) Effects of TRIM71 deficiency on SeV-induced dimerization of IRF3. TRIM71-KO and control HEK293T cells were infected with SeV for 8 h, and then, the cell lysates were separated via native PAGE and analyzed by immunoblotting with the indicated antibodies. (E) Effects of TRIM71 deficiency on the SeV-induced nuclear translocation of IRF3. TRIM71-KO and control HEK293T cells were infected with SeV for 8 h, and then, immunoblot analysis of IRF3 in the cytoplasmic (Cyto) and nuclear fractions was performed with the indicated antibodies. (F) Endogenous immunoprecipitation analysis of the interaction between TRIM71 and IRF3 in HEK293T cells. (G) Endogenous association of TRIM71 with IRF3 and DYRK4. HEK293T cells were infected with SeV for the indicated times. Immunoblotting and immunoprecipitation analysis were performed with the indicated antibodies. (H) Immunoblot analysis of the protein level of IRF3 in HEK293T cells transfected with different amounts of Flag-IRF3 or TRIM71 at different dosages. (I) qPCR analysis of the mRNA level of IRF3 in HEK293T cells transfected with control or TRIM71 for 24 h and then infected with SeV for 8 h. (ns: not significant) (*n* = 3 biological replicates). (J) Immunoblot analysis of the protein level of IRF3 in HEK293T cells infected with SeV for the indicated times. (K) Immunoblot analysis of the protein level of IRF3 in HEK293T cells transfected with Flag-IRF3 and its mutants for 20 h, preinfected with SeV for 12 h, and then treated with CHX for the indicated times. (L) Immunoblot analysis of the protein level of IRF3 in HEK293T cells transfected with Flag-IRF3 and its mutants (IRF3-2A, IRF3-5D) together with a control and TRIM71 for 24 h. Data information: Data are representative of three biological replicates and are shown as the means with SEMs (B, I); data in (A, C–H, J–L) are representative of two replicates. *P < 0.05, ***P < 0.001; two-tailed unpaired Student's *t* test.

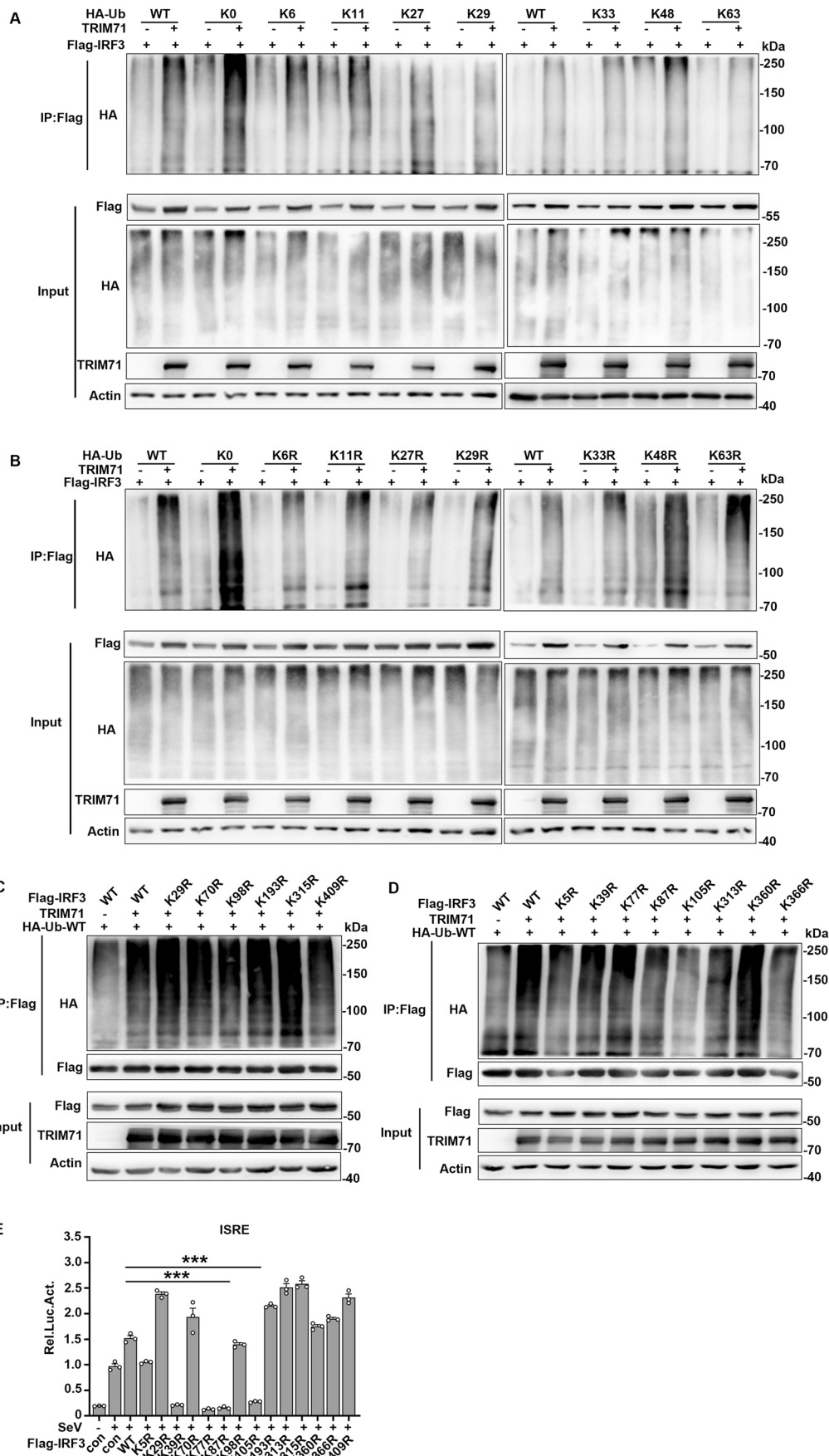

◀ **Figure EV5.    TRIM71 catalyzes the K0-linked ubiquitination of IRF3 at Lys87 and Lys105.**

(A, B) Coimmunoprecipitation analysis of the ubiquitination of IRF3 in HEK293T cells transfected with plasmids encoding Flag-IRF3, TRIM7, HA-Ub (WT) and its mutants for 24 h. (K48 cells were treated with MG132 for 4 h before sample collection). (C, D) Coimmunoprecipitation analysis of the ubiquitination of IRF3 in HEK293T cells transfected with plasmids encoding Flag-IRF or its mutants, TRIM71 and HA-Ub (WT), for 24 h. (E) Effects of IRF3 and its mutants on SeV-induced activation of ISRE. HEK293T cells were transfected with the ISRE reporter, Flag-IRF3, or its mutant plasmids for 24 h and then infected with SeV for 12 h before luciferase assays. (P value; K87R: $1.2 \times 10^{-5}$, K105R: $1.6 \times 10^{-5}$) ($n = 3$ biological replicates). Data information: Data are representative of three biological replicates and are shown as the mean with SEM (E); data in (A–D) are representative of two replicates. ***$P < 0.001$, two-tailed unpaired Student's $t$ test.

