## [Peer Review File · EMBO Reports]

DYRK4 upregulates antiviral innate immunity by promoting IRF3 activation

Xianhuang Zeng, Jiaqi Xu, Jiaqi Liu, Yang Liu, Siqi Yang, Junsong Huang, Chengpeng Fan, Mingxiong Guo, and Guihong Sun

Corresponding author(s): Guihong Sun (ghsunlab@whu.edu.cn) , Mingxiong Guo (guomx@whu.edu.cn)

Review Timeline:

Submission Date:	14th Feb 24
Editorial Decision:	20th Mar 24
Revision Received:	20th Sep 24
Editorial Decision:	28th Oct 24
Revision Received:	10th Nov 24
Accepted:	27th Nov 24

Editor: Achim Breiling

Transaction Report:

Dear Prof. Guo

Thank you for the submission of your manuscript to EMBO reports. I have now received the reports from the three referees that were asked to evaluate your study, which can be found at the end of this email.

As you will see, all referees have several comments, concerns, and suggestions, indicating that a major revision of the manuscript is necessary to allow publication of the study in EMBO reports. As the reports are below, and all the concerns need to be addressed, I will not detail them here.

Given the constructive referee comments, I would like to invite you to revise your manuscript with the understanding that the concerns of the referees must be addressed in the revised manuscript or in a detailed point-by-point response. Acceptance of your manuscript will depend on a positive outcome of a second round of review. It is EMBO reports policy to allow a single round of revision only and acceptance of the manuscript will therefore depend on the completeness of your responses included in the next, final version of the manuscript.

- 1) a .docx formatted version of the final manuscript text (including legends for main figures, EV figures and tables), but without the figures included. Figure legends should be compiled at the end of the manuscript text.
- 2) individual production quality figure files as .eps, .tif, .jpg (one file per figure), of main figures and EV figures. Please upload these as separate, individual files upon re-submission.

- 4) a complete author checklist, which you can download from our author guidelines (<https://www.embopress.org/page/journal/14693178/authorguide>). Please insert page numbers in the checklist to indicate where the requested information can be found in the manuscript. The completed author checklist will also be part of the RPF.

- 5) that primary datasets produced in this study (e.g. RNA-seq, ChIP-seq, structural and array data) are deposited in an

appropriate public database. If no primary datasets have been deposited, please also state this in a dedicated section (e.g. 'No primary datasets have been generated and deposited'), see below.

The accession numbers and database should be listed in a formal "Data Availability" section (placed after Materials & Methods) that follows the model below. This is now mandatory (like the COI statement). Please note that the Data Availability Section is restricted to new primary data that are part of this study. This section is mandatory. As indicated above, if no primary datasets have been deposited, please state this in this section

Data availability

8) Regarding data quantification and statistics, please make sure that the number "n" for how many independent experiments were performed, their nature (biological versus technical replicates), the bars and error bars (e.g. SEM, SD) and the test used to calculate p-values is indicated in the respective figure legends (also for EV figures and all those in an Appendix). Please also check that all the p-values are explained in the legend, and that these fit to those shown in the figure. Please provide statistical testing where applicable. Please avoid the phrase 'independent experiment', but clearly state if these were biological or technical replicates. Please also indicate (e.g. with n.s.) if testing was performed, but the differences are not significant. In case n=2, please show the data as separate datapoints without error bars and statistics. See also: <http://www.embopress.org/page/journal/14693178/authorguide#statisticalanalysis>

9) Please add scale bars of similar style and thickness to microscopic images, using clearly visible black or white bars (depending on the background). Please place these in the lower right corner of the images themselves. Please do not write on or near the bars in the image but define the size in the respective figure legend.

10) Please also note our reference format:

12) We now use CRedit to specify the contributions of each author in the journal submission system. CRedit replaces the author contribution section. Please use the free text box to provide more detailed descriptions and do not provide your final manuscript text file with an author contributions section. See also our guide to authors: <https://www.embopress.org/page/journal/14693178/authorguide#authorshipguidelines>

13) We would encourage you to use 'Structured Methods', our new Materials and Methods format. According to this format, the

Materials and Methods section should include a Reagents and Tools Table (listing key reagents, experimental models, software, and relevant equipment and including their sources and relevant identifiers), uploaded as separate file, followed by a Methods and Protocols section in which we encourage the authors to describe their methods using a step-by-step protocol format with bullet points, to facilitate the adoption of the methodologies across labs. More information on how to adhere to this format as well as downloadable templates (.doc or .xls) for the Reagents and Tools Table can be found in our author guidelines (section 'Structured Methods'):

14) Please add up to five keywords to the manuscript and provide the abstract written in present tense. Please also order the manuscript sections like this, using these names:

Title page - Abstract - Keywords - Introduction - Results - Discussion - Materials and Methods - Data availability section - Acknowledgements - Disclosure and Competing Interests Statement - References - Figure legends - Expanded View Figure legends

Finally, please note that all corresponding authors are required to supply an ORCID ID for their name upon submission of a revised manuscript. Presently, an ORCID is missing for co-corresponding author Guihong Sun. Please find instructions on how to link the ORCID ID to the account in our manuscript tracking system in our Author guidelines:

<http://www.embopress.org/page/journal/14693178/authorguide#authorshipguidelines>

I look forward to seeing a revised version of your manuscript when it is ready. Please let me know if you have questions or comments regarding the revision.

Yours sincerely,

Referee #1:

This manuscript reported a non-enzymatic function of DYRK4. This enzyme acts as a scaffold protein that brings LUBAC and TRIM71 to IRF3 and facilitates its linear ubiquitylation and activation to protect it from degradation upon viral infection. DYRK4 ubiquitylates IRF3, which forms a homodimer and gets translocated into the nucleus which further acts as a transcription factor to upregulate anti-viral response genes such as IFNs, ISRE, NFkB, etc. The biochemical experiments in cells and the experiments done in DYRK4 KO mice concur with each other. Overall the work is of interest for kinase-ubiquitin and antiviral response field. However, several points need to be addressed before the manuscript can be considered for publication.

Following points need to be addressed:

- DYRK2 is also shown to be involved in regulation of antiviral responses. Now that the authors show DYRK4 has role in this process, is the antiviral response a redundant function of DYRK2/DYRK4 or they functioning in independent pathways
- The authors need to test other DYRK family members for IRF3 activation and antiviral responses to demonstrate the specificity of DYRK4 or otherwise.
- Is Trim71 and LUBAC association specific to DYRK4 or interact with other DYRK members as well?
- Interaction between IRF3 and Trim71 needs to be tested in vitro. And, whether presence of DYRK4 in these in vitro experiments affect TRIM71 and IRF3 interaction.
- The data on LUBAC involvement for IRF3 activation is not well developed and is not very clear. More data is needed to support LUBAC involvement in IRF3 stability and subsequent anti-viral response.
- According to authors, IF LUBAC is mediating linear ubiquitination and stabilizing IRF3, why is TRIM71 catalytic E3 ligase activity required for IRF3 ubiquitination?
- Is TRIM17 also acting as an adaptor for LUBAC recruitment to IRF3? Does LUBAC components bind to IRF3. Also, will this interaction be dependent on Trim71 and/or DYRK4?
- The authors have suggested that "DYRK4 mediates K0 linear ubiquitination of IRF3 through TRIM71 and is associated with the LUBAC complex, indicating that DYRK4, TRIM71, and LUBAC may form a large complex to synergistically mediate K0 linear ubiquitination of IRF3." However, the experiments showed that it's a mixed chain type Ub linkage. To claim this different types of Ub linkages need to be checked in TRIM KO and DYRK KO conditions where K0 linkage will be unaltered upon virus infection and all other linkages remain unaffected. Fig: 7D does not reflect the claim.
- The ring domain of TRIM71 is required for binding to DYRK4 and IRF3, where DYRK4 acts as a scaffold. This experiment needs to be done in in-vitro condition as well as in TRIM71 KO condition by over-expressing only the RING domain of TRIM71 for validation. In Fig: 7G, DYRK4 KO#2 shows significant binding. Whether the stability of IRF3 in TRIM71 KO can be rescued

only by expressing the Ring domain of TRIM71 needs to be checked.

- What is the fate of this interaction, Ubiquitylation, and stability of IRF3 transcription factor in cells that are not involved in anti-viral response and interferon production? What are the transcriptional targets of IRF3 in other cell types?
- The stability of K87R, K105R, K87/105R, and half-life has to be shown by cycloheximide assay. The dimerization status and phosphorylation status of these mutants also needs to be checked.
- Fig: 1I, whether TRIM71 & IRF3 KO/KD replicate the same phenotype, needs to be checked.
- Most importantly, all the ubiquitination assays of IRF3 are done in cells. In vitro ubiquitination assays with purified complexes need to be performed to clearly demonstrate that DYRK4 is indeed necessary to recruit TRIM71/LUBAC to ubiquitinate IRF3.

Minor:

- Manuscript contains several typographical and grammatical errors. Authors need to carefully read the manuscript and correct all the errors.
- For instance: "Interestingly, we found that overexpression of DYRK4 promoted K0 (a lysine-free Ub mutant) ubiquitination of IRF3 (Fig. 5G), suggesting that it is the K0 linear ubiquitination of IRF3 that DYRK4 may be affected in" line is incomplete. There are several other errors that need to be corrected.
- Molecular weight markers are not mentioned throughout the figures of the manuscripts. That needs to be incorporated in all the western blot figures.
- Upon SeV infection IRF3 level should go down, which is reflected in other blots but not in Fig: S1E.
- As per Fig: 5G, and Fig: S8A & S8B, various kinds of Ub linkages are formed, though the author has claimed only about K0 linkage in the test. This needs correction.
- Fig: 7C, can be replaced with a better blot.
- Fig S7: loading control and denaturing SDS- PAGE is missing.
- Fig: 7M, the blot needs to be quantified. The difference between WT & CA is not so evident. The same can be replaced with a better image.
- Input of various HA-Ub mutants needs to be added to the figures as their expression pattern might vary due to stability.
- Fig: 7D, the legend is not matching with the figure. The text also needs to be corrected.
- Fig: S7, figure legends & numbering do not match with figures.
- Fig: 5B (left panel), wild-type blot looks like a transfer problem. The same thing (WT & MT) needs to be represented in the same gel.

Referee #2:

In this manuscript, the authors report a novel antiviral signaling pathway mediated by DYRK4, a Ser/Thr kinase, which activates IRF3, a key component of cellular antiviral responses independent of kinase activity. DYRK4 stabilizes IRF3 by working together with TRIM71 and LUBAC, which linearly ubiquitinates IRF3 on specific Lys residues. DYRK4 interacts with TRIM71, which recruits LUBAC to catalyze linear ubiquitination of IRF3. Using complementary cellular and mouse models, the authors demonstrate the requirement of DYRK4 and its kinase-independent function in this pathway. Extensive biochemical and genetic results were provided to dissect the new IRF3 activation mechanism.

Although novel, there are a number of weaknesses, particularly in the rigor of the approach noted below, which should be considered further.

- The effects of DYRK4 on IRF3 are marginal in many cases; the IRF3 activation and gene induction changes are only marginally affected. This could be due to cell-specific differences. It is also unclear why two different KO clones would behave differently.
 - The specificity of DYRK4 is not clarified. DYRK4's effects on NF- κ B are evident from the results, but there are no further follow-up studies to dissect this further.
- The bulk of the studies are based on the HEK293-based cell model, which is fine for mechanistic purposes but not justified for physiological relevance.
- Most of the IRF3 activation studies were performed in cells expressing the endogenous protein, and this is not ideal for studying the biochemical mechanism of the ectopically expressed protein. Ideally, KO cells should be reconstituted with Wt or mutant proteins to demonstrate physiological functions.
 - A major concern of the study is over-reliance on overexpression systems for all the newly identified proteins. Molecular interactions should be validated using endogenous proteins.
 - In some cases, the figure citations are incorrect, e.g., Fig 7D, 7E, etc. Please correct all the citations appropriately.
 - Discussion is too long and can be shortened.

Referee #3:

Interferon regulatory factor 3 (IRF3) is a central transcription factor involved in the innate immune response. Upon activation of pattern recognition receptor such as RLRs, TLR3 or cGAS, a signaling cascade is initiated, resulting in the phosphorylation and translocation of IRF3 into the nucleus where it induces the expression of type I interferons, thereby inducing an antiviral response. Chattopadhyay et al (Immunity, 2016) showed that linear ubiquitination of IRF3 mediated by linear ubiquitin chain assembly complex (LUBAC) activates the RLR-induced IRF3. DYRK4 (Dual-specificity tyrosine-phosphorylation-regulated kinase 4) is a member of the DYRK family of protein kinases, known for their involvement in diverse cellular processes. However, little is known about DYRK4, and it has not yet been linked to innate immune regulation. Here Zeng and co-workers show that DYRK4 upregulates innate immune activation by promoting the linear ubiquitination of IRF3. In line, mice lacking DYRK4 have increased susceptibility to VSV infection and lower cytokine release.

Mechanistically, the authors show that DYRK4 interacts with TRIM71 and IRF3, which promotes TRIM71-mediated linear ubiquitination of IRF3, enhancing IRF3's stability.

Overall, this is an interesting and novel analysis of the role of DYRK4 in innate immunity. The data presented support the suggested mechanism and the experiments are well performed. However, the overall impact of DYRK4 especially on IRF3 levels and ubiquitination seems occasionally modest and the analyses should be strengthened by thorough quantifications. Importantly, the link between DYRK4 and TRIM71 should be improved and the mass spectrometry analyses that led to the identification of TRIM71 shown.

Major:

Some analyses would benefit from quantifications to support the main hypotheses. Please show a quantification based on n=3 independent experiments. Fig. 1 I FACS analysis of percentage of infected cells, pIRF3 levels for Fig. 2 E-I, IRF3 levels for Fig. 5 A-C, Ubiquitination levels Fig. 5 F, 7 F and G.

The involvement of TRIM71 in IRF3 stability has to be strengthened. It is important that endogenous IRF3 needs to be shown, assays with overexpressed IRF3 may lead to misleading results. Fig. 7C is a key experiment, and thus should be repeated to show a clearer endogenous ubiquitination, lower p-IRF3 levels in TRIM71 KO cells. Furthermore, the Ubiquitination should be quantified in three independent repeats.

Fig. 7 J: TRIM71 does not seem to promote IRF3 stability upon infection, please either show this or refrain from stating it. Fig 7 M: TRIM71 -WT and TRIM71-CA seem to have the same effect on IRF3 levels. Please show and analyse the Mass Spectrometry data e.g. in the introduction to Fig. 6.

Linear ubiquitin chains are harder to identify using only mutants. To show that IRF3 is attached to linear ubiquitin chains, the DYRK4/TRIM71 ubiquitinated IRF3 purified from cells should be digested with Otulin or CYLD in vitro (see also Mevissen et al, Cell, 2013).

Minor:

- Fig. 2 E-I: Opposed to the statement in the main text, the shown blots do not show a clearly decreased phosphorylation of IRF3 in DYRK4 KO cells. Quantifications of independent repeats would help.
- Fig. 4 B: The Flag-IRF3 blot is missing
- Fig. 5F could be moved to the supplements, the impact on IRF3 stability is weak.
- Fig. 7 C and D are referred as Fig. 7 D and E in the results section
- Fig. S2 G: The VSV titers are not reflected by the GFP expression levels shown by western blot
- Fig. S3 A: Why do the DYRK4 mutants display substantially elevated luciferase activity compared to the wild-type DYRK4?
- Fig. S5 B: the blots for Flag-DYRK4 are missing in the IP
- kDa markers are incomplete or missing in most Western Blots
- The (almost) same sentence is repeated twice when describing Fig. S8A.
- The introduction and discussion should be streamlined and unnecessary detail removed to improve clarity.

Referee #1:

This manuscript reported a non-enzymatic function of DYRK4. This enzyme acts as a scaffold protein that brings LUBAC and TRIM71 to IRF3 and facilitates its linear ubiquitylation and activation to protect it from degradation upon viral infection. DYRK4 ubiquitylates IRF3, which forms a homodimer and gets translocated into the nucleus which further acts as a transcription factor to upregulate anti-viral response genes such as IFNs, ISRE, NFkB, etc. The biochemical experiments in cells and the experiments done in DYRK4 KO mice concur with each other. Overall the work is of interest for kinase-ubiquitin and antiviral response field. However, several points need to be addressed before the manuscript can be considered for publication.

Following points need to be addressed:

Dear reviewer,

Thank you very much for your comments and professional advice. These opinions help to improve academic rigor of our article. Based on your suggestion and request, we have made corrected modifications on the revised manuscript. As review said that “several points need to be addressed before the manuscript can be considered for publication”, now we focus on solving these problems. Meanwhile, we performed immunoblotting again to improve the quality of the figures and corrected typographical and grammatical errors in the manuscript. We hope that our work can be improved again. Furthermore, we would like to show the detail as follows and all revisions in the manuscript have been highlighted.

- DYRK2 is also shown to be involved in regulation of antiviral responses. Now that the authors show DYRK4 has role in this process, is the antiviral response a redundant function of DYRK2/DYRK4 or they functioning in independent pathways

Response: We thank the reviewer's comment. DYRK2 inhibited the virus-triggered induction of type I IFNs and promoted the K48-linked ubiquitination and degradation of TBK1 in a kinase-activity-dependent manner¹. In our manuscript, we found that DYRK4 promoted the virus-triggered induction of type I IFNs and promoted the K0-linked ubiquitination and stability of IRF3 in a kinase-activity-independent manner. Therefore, the antiviral response of DYRK2/DYRK4 works in independent pathways.

- The authors need to test other DYRK family members for IRF3 activation and antiviral responses to demonstrate the specificity of DYRK4 or otherwise.

Response: Thank you very much for your professional advice, we constructed plasmids for all DYRKs family members and tested their effects on IRF3 activation and antiviral responses through dual-luciferase and western blot experiments (A and B). We found that DYRK2 indeed inhibited the virus-triggered induction of type I IFNs as literature¹, and DYRK4 significantly promotes the virus-triggered induction of type I IFNs and phosphorylation of IRF3.

Figure for referee with unpublished data has been removed upon request by the authors.

- Is Trim71 and LUBAC association specific to DYRK4 or interact with other DYRK members as well?

Response: Done as suggested. We performed Co-immunoprecipitation experiments to analyze the association of TRIM71 and LUBAC (Sharpin) with other DYRKs members. It was shown that only DYRK4 interacts with Sharpin or TRIM71.

Figure for referee with unpublished data has been removed upon request by the authors.

- Interaction between IRF3 and Trim71 needs to be tested in vitro. And, whether presence of DYRK4 in these in vitro experiments affect TRIM71 and IRF3 interaction.

Response: Done as suggested. We performed GST pull-down experiments to analyze the association of TRIM71 with IRF3 in vitro. And DYRK4 indeed promotes the interaction between IRF3 and Trim71.

Figure for referee with unpublished data has been removed upon request by the authors.

- The data on LUBAC involvement for IRF3 activation is not well developed and is not very clear. More data is needed to support LUBAC involvement in IRF3 stability and subsequent anti-viral response.

Response: Done as suggested. We performed qPCR and western blot experiments to analyze the involvement of LUBAC (Sharpin) in IRF3 stability and subsequent antiviral responses. Overexpression of Sharpin enhanced SeV-induced transcription of IFNB1 and SeV-induced phosphorylation of IRF3 (A, B). In addition, under conditions of viral infection, overexpression of Sharpin can extend the half-life of endogenous IRF3 protein (C). And IRF3 stability and activation will can promote the subsequent anti-viral response.

Figure for referee with unpublished data has been removed upon request by the authors.

- According to authors, IF LUBAC is mediating linear ubiquitination and stabilizing IRF3, why is TRIM71 catalytic E3 ligase activity required for IRF3 ubiquitination?

Response: We thank the reviewer for pointing out this issue. Regarding "why is TRIM71 catalytic E3 ligase activity required for IRF3 ubiquitination", as mentioned in the discussion, DYRK4, TRIM71 and LUBAC may form a large complex to cooperatively mediate the K0

linear ubiquitination of IRF3; so, it has an E3 ligase catalytic active TRIM71 may contribute to the formation of DYRK4, TRIM71, and LUBAC complexes. In addition, we also found that the complete enzymatic activity of TRIM71 is necessary for interaction with LUBAC (HOIP), and TRIM71 also acts as an adapter protein for LUBAC (Sharpin) to recruit IRF3, so IRF3 ubiquitination requires the enzymatic activity of TRIM71. But in regard to the specific relationship between LUBAC and TRIM71, this is an interesting topic, which will be the next topic that we need to continue to study in depth.

Figure for referee with unpublished data has been removed upon request by the authors.

- Is TRIM17 also acting as an adaptor for LUBAC recruitment to IRF3? Does LUBAC components bind to IRF3. Also, will this interaction be dependent on Trim71 and/or DYRK4?

Response: We thank the reviewer for pointing out this issue. We also believe that TRIM71 acts as an adaptor for LUBAC recruitment to IRF3, and Co-IP experiments show that TRIM71 promotes the interaction between LUBAC (Sharpin) and IRF3. It has been shown in the literature that LUBAC components interact with IRF3 ², and we confirmed this result. In addition, DYRK4 or TRIM71 indeed enhanced the interaction between LUBAC (Sharpin) and IRF3.

Figure for referee with unpublished data has been removed upon request by the authors.

- The authors have suggested that "DYRK4 mediates K0 linear ubiquitination of IRF3 through TRIM71 and is associated with the LUBAC complex, indicating that DYRK4, TRIM71, and LUBAC may form a large complex to synergistically mediate K0 linear ubiquitination of IRF3." However, the experiments showed that it's a mixed chain type Ub linkage. To claim this different types of Ub linkages need to be checked in TRIM KO and DYRK KO conditions where K0 linkage will be unaltered upon virus infection and all other linkages remain unaffected. Fig: 7D does not reflect the claim.

Response: Done as suggested. We reconstituted TRIM71-KO and DYRK4-KO cell lines, and then examined different types of Ub linkages of IRF3 under TRIM71-KO and DYRK4-KO conditions, and found that K0 linkage was unaltered upon virus infection and all other linkages remain unaffected. Furthermore, in Figure 7D, our idea was to verify that DYRK4 mediates K0-linked ubiquitination of IRF3 through TRIM71 to indicate the importance of TRIM71 in the complex. But in regard to "DYRK4, TRIM71 and LUBAC may form a large complex to cooperatively mediate K0-linked ubiquitination of IRF3", this is also an interesting topic and need to continue to study in depth.

Figure for referee with unpublished data has been removed upon request by the authors.

- The ring domain of TRIM71 is required for binding to DYRK4 and IRF3, where DYRK4 acts as a scaffold. This experiment needs to be done in in-vitro condition as well as in TRIM71 KO condition by over-expressing only the RING domain of TRIM71 for validation. In Fig: 7G, DYRK4 KO#2 shows significant binding. Whether the stability of IRF3 in TRIM71 KO can be rescued only by expressing the Ring domain of TRIM71 needs to be checked.

Response: Done as suggested. We performed Co-IP experiments to analyze the interaction of the RING domain of TRIM71 with DYRK4 and IRF3 under TRIM71-KO conditions. Co-IP experiments show that the RING domain of TRIM71 is required for interaction with DYRK4 or IRF3 (A, B). For "In Fig: 7G (actually referring to Fig: 6G), DYRK4 KO#2 shows significant binding". This is because we used two different sgRNA sequences to target different sites in the genome to construct two DYRK4-KO cell lines. Although this will lead to differences in the DYRK4-KO cell lines, compared with wild-type cells, it can still be seen

that knocking out DYRK4 will weaken the interaction between TRIM71 and IRF3. For "Whether the stability of IRF3 in TRIM71 KO can be rescued only by expressing the Ring domain of TRIM71 needs to be checked." We performed immunoblotting analysis and found that the stability of IRF3 in TRIM71 KO could be rescued by expressing the Ring domain of TRIM71 compared to control.

Figure for referee with unpublished data has been removed upon request by the authors.

- What is the fate of this interaction, Ubiquitylation, and stability of IRF3 transcription factor in cells that are not involved in anti-viral response and interferon production? What are the transcriptional targets of IRF3 in other cell types?

Response: We thank the reviewer for this interesting question. We speculate that, when ubiquitylation and stability of IRF3 transcription factor are not involved in anti-viral response and interferon production, the stability of IRF3 may also affects cell fate, such as a recent literature report that IRF3 activates RB to authorize cGAS-STING-induced senescence and mitigate liver fibrosis ³. The authors found that C-terminally phosphorylated IRF3 forms an endogenous complex with the tumor suppressor RB in the nucleus, thereby disrupting the CDK4/6-cyclin-RB complex, thereby maintaining RB in a hypophosphorylated and active state. And another literature reported that Hepatic IRF3 fuels dysglycemia in obesity through direct regulation of Ppp2r1b ⁴.

- The stability of K87R, K105R, K87/105R, and half-life has to be shown by cycloheximide assay. The dimerization status and phosphorylation status of these mutants also needs to be checked.

Response: Done as suggested. We tested the stability of K87R and half-life by cycloheximide assay (A). Actually, as shown in Fig. S7K we have shown the stability of K87/105R and half-life by cycloheximide assay. In addition, we also detected the dimerization status and phosphorylation status of the K87/105R. The results showed that the dimerization and phosphorylation of the K87/105R were inhibited compared to control (B, C and D). (Note: Overexpression of wild-type or mutant IRF3 in the resting state can lead to a small amount of dimer formation).

Figure for referee with unpublished data has been removed upon request by the authors.

- Fig: 1I, whether TRIM71 & IRF3 KO/KD replicate the same phenotype, needs to be checked.

Response: Done as suggested. We first constructed IRF3-KD and TRIM71-KD A549 cell lines and then infected the cells with VSV-GFP. The results showed that IRF3-KD and TRIM71-KD replicated the same phenotype as in Fig.1I.

Figure for referee with unpublished data has been removed upon request by the authors.

- Most importantly, all the ubiquitination assays of IRF3 are done in cells. In vitro ubiquitination assays with purified complexes need to be performed to clearly demonstrate that DYRK4 is indeed necessary to recruit TRIM71/LUBAC to ubiquitinate IRF3.

Response: Done as suggested. For in vitro ubiquitination experiments, we performed an in vitro ubiquitination assay using purified proteins to analyze ubiquitin-modified IRF3 and found that DYRK4 is indeed necessary to recruit TRIM71/LUBAC to ubiquitinate IRF3.

Figure for referee with unpublished data has been removed upon request by the authors.

Minor:

- Manuscript contains several typographical and grammatical errors. Authors need to carefully read the manuscript and correct all the errors.

Response: We thank the reviewer for pointing out this issue. Done as suggested.

- For instance: "Interestingly, we found that overexpression of DYRK4 promoted K0 (a lysine-free Ub mutant) ubiquitination of IRF3 (Fig. 5G), suggesting that it is the K0 linear ubiquitination of IRF3 that DYRK4 may be affected in" line is incomplete. There are several other errors that need to be corrected.

Response: Done as suggested.

- Molecular weight markers are not mentioned throughout the figures of the manuscripts. That needs to be incorporated in all the western blot figures.

Response: Done as suggested.

- Upon SeV infection IRF3 level should go down, which is reflected in other blots but not in Fig: S1E.

Response: Done as suggested. Fig: S1E, has been replaced with a better blot.

- As per Fig: 5G, and Fig: S8A & S8B, various kinds of Ub linkages are formed, though the author has claimed only about K0 linkage in the text. This needs correction.

Response: Done as suggested.

- Fig: 7C, can be replaced with a better blot.

Response: Done as suggested. Fig: 7C, has been replaced with a better blot.

- **Fig S7: loading control and denaturing SDS- PAGE is missing.**

Response: Done as suggested. We supplemented Fig: S7 with the missing loading control and denaturing SDS-PAGE.

- **Fig: 7M, the blot needs to be quantified. The difference between WT & CA is not so evident. The same can be replaced with a better image.**

Response: Done as suggested. Fig: 7M, has been replaced with a better blot.

- **Input of various HA-Ub mutants needs to be added to the figures as their expression pattern might vary due to stability.**

Response: Done as suggested. We supplemented the Input of various HA-Ub mutants to the figures.

- **Fig: 7D, the legend is not matching with the figure. The text also needs to be corrected.**

Response: We thank the reviewer for pointing out this issue. Done as suggested. We have corrected all citations.

- **Fig: S7, figure legends & numbering do not match with figures.**

Response: Done as suggested.

- **Fig: 5B (left panel), wild-type blot looks like a transfer problem. The same thing (WT& MT) needs to be represented in the same gel.**

Response: Fig: 5B (left panel), has been replaced with a better blot. In order to observe the real effect more intuitively, we enhance the analyses by quantification. Quantification of relative IRF3 levels in DYRK4-WT and MT groups is shown on the right. It was shown that overexpression of DYRK4-WT or MT stabilized endogenous IRF3 protein (Fig. 5B).

Figure for referee with unpublished data has been removed upon request by the authors.

Referee #2:

In this manuscript, the authors report a novel antiviral signaling pathway mediated by DYRK4, a Ser/Thr kinase, which activates IRF3, a key component of cellular antiviral responses independent of kinase activity. DYRK4 stabilizes IRF3 by working together with TRIM71 and LUBAC, which linearly ubiquitinates IRF3 on specific Lys residues. DYRK4 interacts with TRIM71, which recruits LUBAC to catalyze linear ubiquitination of IRF3. Using complementary cellular and mouse models, the authors demonstrate the requirement of DYRK4 and its kinase-independent function in this pathway. Extensive biochemical and genetic results were provided to dissect the new IRF3 activation mechanism.

Response:

Dear reviewer,

Thank you very much for your comments and professional advice. These opinions help to improve academic rigor of our article. Based on your suggestion and comments, we have made corrected modifications on the revised manuscript. As review said that “there are a number of weaknesses, particularly in the rigor of the approach noted below, which should be considered further”, we have made improvements to the rigor of the approach noted below. Furthermore, we reconstituted the IRF3-KO cells using Wt or mutant proteins to demonstrate physiological functions. Meanwhile, the discussion has also been streamlined and unnecessary details removed to improve clarity. We hope that our work can be improved again. Furthermore, we would like to show the detail as follows and all revisions in the manuscript have been highlighted.

- The effects of DYRK4 on IRF3 are marginal in many cases; the IRF3 activation and gene induction changes are only marginally affected. This could be due to cell-specific differences. It is also unclear why two different KO clones would behave differently.

Response: We thanks reviewer for the professional comments. Because we used two different sgRNA sequences to target different sites in the genome and constructed two 293T or A549 DYRK4-KO/KD cell lines, different sgRNA sequences may cause cell-specific differences. The two different KO/KD clones basically showed the similar phenotype in IRF3 activation and gene induction changes, for example, Fig: 1C, 1D, 1E, 1G and Fig: S2B, S2C, S2E. So, the effect of DYRK4 on IRF3 activation and gene induction changes is quite obvious and consistent.

- The specificity of DYRK4 is not clarified. DYRK4's effects on NF-kB are evident from the results, but there are no further follow-up studies to dissect this further.

The bulk of the studies are based on the HEK293-based cell model, which is fine for mechanistic purposes but not justified for physiological relevance.

Response:

We thank the reviewer for the professional comments. From the results that DYRK4 affects NF-kB signaling and that DYRK4 can promote virus-induced NF-kB pathway activation. In this study, the topic of our study is the effect on IRF3 activation and stability, and furthermore the effects on anti-viral response. However, further clarifying the specific effect of DYRK4 on NF-kB will be the next topic that we need to continue to study in depth.

Regarding the physiological relevance, we conducted studies at the cellular and animal levels.

At the cellular level, we used human cell lines 293T (Fig: 1A, 1B, 1C, 1E, 1H, 1G and Fig: S1, S2, S3), A549 (Fig: 1D, 1F, 1I and Fig: S2), and mouse cell line RAW264.7 (Fig: S3D), mouse primary cells BMDM, BMDC, MLF (Fig: 2). In addition, there is a DYRK4 knockout mouse model (Fig: 3). So, we think these study models are justified for physiological relevance.

- Most of the IRF3 activation studies were performed in cells expressing the endogenous protein, and this is not ideal for studying the biochemical mechanism of the ectopically expressed protein. Ideally, KO cells should be reconstituted with Wt or mutant proteins to demonstrate physiological functions.

Response: Done as suggested. We thank the reviewer for the professional suggestions. We have constructed an IRF3-KO cell line (A), and detected the dimerization status and phosphorylation status of the IRF3-WT or K87/105R. The results showed that the dimerization and phosphorylation of the K87/105R were inhibited compared to control (B, C). (Note: Overexpression of wild-type or mutant IRF3 in the resting state can lead to a small amount of dimer formation).

Figure for referee with unpublished data has been removed upon request by the authors.

- A major concern of the study is over-reliance on overexpression systems for all the newly identified proteins. Molecular interactions should be validated using endogenous proteins.

Response: We thank the reviewer for the professional comments. Actually, we have used endogenous proteins to verify molecular interactions as shown in Fig. 4E and Fig. S7F, S7G.

- In some cases, the figure citations are incorrect, e.g., Fig 7D, 7E, etc. Please correct all the citations appropriately.

Response: We thank the reviewer for pointing out this issue. Done as suggested. We have corrected all citations.

- Discussion is too long and can be shortened.

Response: Done as suggested.

Referee #3:

Interferon regulatory factor 3 (IRF3) is a central transcription factor involved in the innate immune response. Upon activation of pattern recognition receptor such as RLRs, TLR3 or cGAS, a signaling cascade is initiated, resulting in the phosphorylation and translocation of IRF3 into the nucleus where it induces the expression of type I interferons, thereby inducing an antiviral response. Chattopadhyay et al (Immunity, 2016) showed that linear ubiquitination of IRF3 mediated by linear ubiquitin chain assembly complex (LUBAC) activates the RLR-induced IRF3. DYRK4 (Dual-specificity tyrosine-phosphorylation-regulated kinase 4) is a member of the DYRK family of protein kinases, known for their involvement in diverse cellular processes. However, little is known about DYRK4, and it has not yet been linked to innate immune regulation

Here Zeng and co-workers show that DYRK4 upregulates innate immune activation by promoting the linear ubiquitination of IRF3. In line, mice lacking DYRK4 have increased susceptibility to VSV infection and lower cytokine release.

Mechanistically, the authors show that DYRK4 interacts with TRIM71 and IRF3, which promotes TRIM71-mediated linear ubiquitination of IRF3, enhancing IRF3's stability.

Overall, this is an interesting and novel analysis of the role of DYRK4 in innate immunity. The data presented support the suggested mechanism and the experiments are well performed. However, the overall impact of DYRK4 especially on IRF3 levels and ubiquitination seems occasionally modest and the analyses should be strengthened by thorough quantifications. Importantly, the link between DYRK4 and TRIM71 should be improved and the mass spectrometry analyses that led to the identification of TRIM71 shown.

Response:

Dear reviewer,

Thank you very much for your comments and professional advice. These opinions help to improve academic rigor of our article. Based on your professional suggestions and comments, we have made corrected modifications on the revised manuscript. As review said that “the overall impact of DYRK4 especially on IRF3 levels and ubiquitination seems occasionally modest and the analyses should be strengthened by thorough quantifications”, we again performed immunoblotting to analyze IRF3 levels and ubiquitination and strengthened the analysis by quantification. Furthermore, we strengthened the link between DYRK4 and TRIM71 by performing in vitro deubiquitination assays and uploading and analyzing Mass Spectrometry data. Meanwhile, the introduction and discussion have also been streamlined and unnecessary details removed to improve clarity. Some analyzes also show a quantitation based on $n = 3$ independent experiments. We hope that our work can be improved again. Furthermore, we would like to show the detail as follows and all revisions in the manuscript have been highlighted.

Major:

Some analyses would benefit from quantifications to support the main hypotheses. Please show a quantification based on $n=3$ independent experiments. Fig. 1 I FACS analysis of percentage of infected cells, pIRF3 levels for Fig. 2 E-I, IRF3 levels for Fig. 5 A-C,

Ubiquitination levels Fig. 5 F, 7 F and G.

Response: Done as suggested. In order to observe the real effect more intuitively, we used ImageJ software for cell counting to quantitatively analyze the percentage of infected cells to show their real effect as shown on the right (Fig. 1I).

Figure for referee with unpublished data has been removed upon request by the authors.

In order to observe the real effect more intuitively, we enhance the analyses by quantification. Quantitation of the relative p-IRF3 levels is shown on the right (Fig. 2E-I).

Figure for referee with unpublished data has been removed upon request by the authors.

In order to observe the real effect more intuitively, we enhance the analyses by quantification. Quantitation of the relative IRF3 levels is shown on the right (Fig. 5A-C).

Figure for referee with unpublished data has been removed upon request by the authors.

In order to observe the real effect more intuitively, we enhance the analyses by quantification. Quantitation of the relative Ubiquitination levels is shown on the right (Fig. 5 F).

Figure for referee with unpublished data has been removed upon request by the authors.

In order to observe the real effect more intuitively, we enhance the analyses by quantification. Quantitation of the relative Ubiquitination levels is shown on the right (Fig. 7F and G)

Figure for referee with unpublished data has been removed upon request by the authors.

The involvement of TRIM71 in IRF3 stability has to be strengthened. It is important that

endogenous IRF3 needs to be shown, assays with overexpressed IRF3 may lead to misleading results. Fig. 7C is a key experiment, and thus should be repeated to show a clearer endogenous ubiquitination, lower p-IRF3 levels in TRIM71 KO cells. Furthermore, the Ubiquitination should be quantified in three independent repeats.

Response: We thank the reviewer for the professional comments. Actually, we have examined the involvement of TRIM71 in the stability of endogenous IRF3 as shown in Fig. 7J, 7L, 7M. Done as suggested. Fig. 7C is indeed a key experiment, and we have adjusted Figure 7C to Figure 7E and revised the text, and Fig. 7E, has been replaced with a better blot to show a clearer endogenous ubiquitination, lower p-IRF3 levels in TRIM71 KO cells. In order to observe the real effect more intuitively, we enhance the analyses by quantification. Quantitation of the relative Ubiquitination levels is shown on the right (Fig. 7E).

Figure for referee with unpublished data has been removed upon request by the authors.

Fig. 7 J: TRIM71 does not seem to promote IRF3 stability upon infection, please either show this or refrain from stating it. Fig 7 M: TRIM71 -WT and TRIM71-CA seem to have the same effect on IRF3 levels. Please show and analyse the Mass Spectrometry data e.g. in the introduction to Fig. 6.

Response: For “Fig. 7J: TRIM71 does not seem to promote IRF3 stability upon infection”. In fact, Wu et al. confirmed that viral infection causes degradation of IRF3⁵. This is reflected in the fact that in the control group of Figure 7J, the IRF3 level at 12h and 24h of virus infection was lower than that in the uninfected state, while the IRF3 level at 12h and 24h of TRIM71 overexpression was almost unchanged compared to that in the uninfected state. In order to observe the real effect more intuitively, we enhance the analyses by quantification. Quantitation of the relative IRF3 levels is shown on the right (Fig. 7J).

Figure for referee with unpublished data has been removed upon request by the authors.

For “Fig 7 M: TRIM71 -WT and TRIM71-CA seem to have the same effect on IRF3 levels.” We again performed immunoblotting to analyze the effects of TRIM71-WT and TRIM71-CA on IRF3 levels, and Fig: 7M, has been replaced with a better blot. As shown below (Fig. 7M).

Figure for referee with unpublished data has been removed upon request by the authors.

For “the Mass Spectrometry data.” We apologize for not displaying and analyzing the Mass Spectrometry data in time. We have uploaded and analyzed the Mass Spectrometry data now. As shown below and in Fig. 6A (on the right).

Figure for referee with unpublished data has been removed upon request by the authors.

Linear ubiquitin chains are harder to identify using only mutants. To show that IRF3 is attached to linear ubiquitin chains, the DYRK4/TRIM71 ubiquitinated IRF3 purified from cells should be digested with Otulin or CYLD in vitro (see also Mevissen et al, Cell, 2013).

Response: We thank the reviewer for the professional comments. To show that IRF3 is attached to linear ubiquitin chains, we purified DYRK4/TRIM71-ubiquitinated IRF3 from cells and digested it with Otulin in vitro for in vitro deubiquitination analysis ⁶, as shown below. The results showed that ubiquitinated IRF3 could be digested by Otulin in vitro, indicating that IRF3 was attached to linear ubiquitin chains.

Figure for referee with unpublished data has been removed upon request by the authors.

Minor:

• **Fig. 2 E-I: Opposed to the statement in the main text, the shown blots do not show a clearly decreased phosphorylation of IRF3 in DYRK4 KO cells. Quantifications of independent repeats would help.**

Response: We thank reviewer for the professional comments. Although IRF3 phosphorylation was reduced in most DYRK4-KO cells in Fig. 2E-I, the reduction was not significant. Now in the revised Fig. 2E-I, we has been replaced with a better blot. In order to observe the real effect more intuitively, we enhance the analyses by quantification. Quantitation of the relative p-IRF3 level is shown on the right (Fig. 2E-I).

Figure for referee with unpublished data has been removed upon request by the authors.

- **Fig. 4 B: The Flag-IRF3 blot is missing**

Response: Done as suggested. We complemented the missing Flag-IRF3 blot.

- **Fig. 5F could be moved to the supplements, the impact on IRF3 stability is weak.**

Response: Done as suggested. Figure 5F has been moved to Supplementary Fig. S5F.

- **Fig. 7 C and D are referred as Fig. 7 D and E in the results section**

Response: We thank the reviewer for pointing out this issue. Done as suggested. We have corrected all citations.

- **Fig. S2 G: The VSV titers are not reflected by the GFP expression levels shown by western blot**

Response: Done as suggested. We have removed the GFP blot.

- **Fig. S3 A: Why do the DYRK4 mutants display substantially elevated luciferase activity compared to the wild-type DYRK4?**

Response: For “Figure S3A” DYRK4 mutants showed significantly elevated luciferase activity. We think there are two reasons: first, although DYRK4 functions independently of kinase activity, the mutant may affect the structure of DYRK4 because the mutation location of the mutant is in the K domain, and the K domain plays a major role. This effect may promote the function of DYRK4. Second, DYRK4 mutants promoted interaction with IRF3 (Fig. 4J) and IRF3 ubiquitination (Fig. 5E) compared to DYRK4-WT.

• **Fig. S5 B: the blots for Flag-DYRK4 are missing in the IP**

Response: Done as suggested.

• **kDa markers are incomplete or missing in most Western Blots**

Response: Done as suggested.

• **The (almost) same sentence is repeated twice when describing Fig. S8A.**

Response: We thank the reviewer for pointing out this issue. We have reworked the sentence.

• **The introduction and discussion should be streamlined and unnecessary detail removed to improve clarity.**

Response: Done as suggested.

References

1. An, T. et al. DYRK2 Negatively Regulates Type I Interferon Induction by Promoting TBK1 Degradation via Ser527 Phosphorylation. *PLoS pathogens* **11**, e1005179 (2015).
2. Chattopadhyay, S., Kuzmanovic, T., Zhang, Y., Wetzel, J.L. & Sen, G.C. Ubiquitination of the Transcription Factor IRF-3 Activates RIPA, the Apoptotic Pathway that Protects Mice from Viral Pathogenesis. *Immunity* **44**, 1151-1161 (2016).
3. Wu, Q. et al. IRF3 activates RB to authorize cGAS-STING-induced senescence and mitigate liver fibrosis. *Science advances* **10**, eadj2102 (2024).
4. Patel, S.J. et al. Hepatic IRF3 fuels dysglycemia in obesity through direct regulation of Ppp2r1b. *Science translational medicine* **14**, eabh3831 (2022).
5. Wu, Y. et al. Selective autophagy controls the stability of transcription factor IRF3 to balance type I interferon production and immune suppression. *Autophagy* **17**, 1379-1392 (2021).
6. Mevissen, T.E. et al. OTU deubiquitinases reveal mechanisms of linkage specificity and enable ubiquitin chain restriction analysis. *Cell* **154**, 169-184 (2013).

Dear Prof. Sun,

Thank you for the submission of your revised manuscript to our editorial offices. I have now received the reports from the referees that I asked to re-evaluate the study, you will find below. As you will see, all three referees now support publication of the study in EMBO reports.

Before I can proceed with formal acceptance, I have these editorial requests I ask you to address in a final revised manuscript:

- Please provide the abstract written in present tense throughout.

- Please order the manuscript sections like this, using these names:

Title page - Abstract - Keywords - Introduction - Results - Discussion - Methods - Data availability section - Acknowledgements (including funding information) - Disclosure and Competing Interests Statement - References - Figure legends - Expanded View Figure legends

- We now use CRediT to specify the contributions of each author in the journal submission system. CRediT replaces the author contribution section. Please use the free text box to provide more detailed descriptions and do NOT provide your final manuscript text file with an author contributions section. See also our guide to authors: <https://www.embopress.org/page/journal/14693178/authorguide#authorshipguidelines>

- We updated our journal's competing interests policy in January 2022 and request authors to consider both actual and perceived competing interests. Please review the policy <https://www.embopress.org/competing-interests> and update your competing interests if necessary. Please name this section 'Disclosure and Competing Interests Statement' and put it after the Acknowledgements section.

- Please provide a complete author checklist with your final submission, which you can download from our author guidelines (<https://www.embopress.org/page/journal/14693178/authorguide>). Please insert page numbers in the checklist to indicate where the requested information can be found in the manuscript. The completed author checklist will also be part of the RPF.

Please also follow our guidelines for the use of living organisms, and the respective reporting guidelines: <http://www.embopress.org/page/journal/14693178/authorguide#livingorganisms>

- Please make sure that for the final submission individual production quality figure files as .eps, .tif, .jpg (one file per figure), of main figures and EV figures are provided. Please upload these as separate, individual files.

- Please add scale bars of similar style and thickness to all microscopic images, using clearly visible black or white bars (depending on the background). Please place these in the lower right corner of the images themselves. Please do not write on or near the bars in the image but define the size in the respective figure legend.

- Please use our reference format:

- Please name the EV figures according to the nomenclature 'Figure EVx' and adjust their legend and callouts accordingly.

- Please make sure that the number "n" for how many independent experiments were performed, their nature (biological versus technical replicates), the bars and error bars (e.g. SEM, SD) and the test used to calculate p-values is indicated in the respective figure legends (main, EV and Appendix figures). Please also check that all the p-values are explained in the legend, and that these fit to those shown in the figure. Please provide statistical testing where applicable. Please avoid the phrase 'independent experiment', but clearly state if these were biological or technical replicates. The standard sentence that is present in all the legends of the manuscript - 'The data represent at least two experiments with similar results (means {plus minus} SDs, n = 3 biological replicates) - is rather unclear. Please indicate for each panel how many replicates were done (where applicable).

Please also indicate (e.g. with n.s.) if testing was performed, but the differences are not significant. In case n=2, please show the data as separate datapoints without error bars and statistics. See also:

<http://www.embopress.org/page/journal/14693178/authorguide#statisticalanalysis>

If n<5, please show single datapoints for diagrams. It seems that presently many diagrams have only partial stats or the 'ns' is missing. Please also place the asterisks indicating significance (p-values) more clearly so it is possible to see which bars are compared. Moreover:

- Please note that the exact p values are not provided in the legends of figures 1A-D; 2A-D; 3B-D; 4A; 6A, B, H, I; 7B, D, H, N; S1 A-C, G, I, K, L; S2 A, C-F, I, L; S3 D, E, G; S4 B; S5 E.

- Please note that information related to n is missing in the legends of figures 5D, S1 G, S2 I, L.

- Please note that the error bars are not defined in the legends of figures 5D.
- Please note that the scale bar is missing for figures 1I; 4D.
- Please note that axis gaps are not labeled appropriately in figures 4A, S1 I.
- Please add to each legend (main, EV figures, where applicable) a 'Data Information' section explaining the statistics used or providing information regarding replicates and scales. See:

- All Materials and Methods need to be described in the main text using our 'Structured Methods' format, which is required for all research articles. According to this format, the Methods section should include a Reagents and Tools Table (listing key reagents, experimental models, software, and relevant equipment and including their sources and relevant identifiers), uploaded as separate file, followed by a Methods section in which we encourage the authors to describe their methods using a step-by-step protocol format with bullet points, to facilitate the adoption of the methodologies across labs. More information on how to adhere to this format as well as downloadable templates (.doc) for the Reagents and Tools Table can be found in our author guidelines (section 'Structured Methods'):

Please add the information provided in the Appendix Tables S1-3 to the Reagents and tools table and update their callouts (e.g. 'see reagents and tools table').

- Appendix Table S4 is a dataset. Please upload this as dataset file named 'Dataset EV1', with a legend on the first TAB of the excel file. Please also add a callout for this dataset to the manuscript main text.
- Please remove the legend for the Appendix Tables from the manuscript text file. Please also remove the Appendix file, as this is now redundant (as the relevant information is provided in the Reagents and tools table and in Dataset EV1).
- The Data Availability section (DAS) is restricted to datasets generated in this study and deposited. If no such datasets have been generated or deposited, please state this here and remove all other information.
- Thank you for providing the requested source data. Please upload this as one folder per figure (with all files for one figure in one folder and ZIPed together).
- Please have the final manuscript text carefully proofread by a native speaker.

In addition, I would need from you uploaded separately:

Best,
 Achim Breiling
 Senior Editor
 EMBO Reports

 Referee #1:

All the reviewer's queries have been addressed satisfactorily.

 Referee #2:

The authors addressed my previous concerns. I would have liked to see some specificity for the mechanism, but the NF-kB

regulation part may be out of scope.

Referee #3:

The author's have sufficiently addressed all my comments and strengthens the manuscript significantly. I have no further concerns.

All editorial and formatting issues were resolved by the authors.

Prof. Guihong Sun
Wuhan University
185 Donghu Road
Wuhan, Hubei 430071
China

Dear Prof. Sun,

I am very pleased to accept your manuscript for publication in the next available issue of EMBO reports. Thank you for your contribution to our journal.

Yours sincerely,
